# A Generalized Label Shift Perspective for Cross-Domain Gaze Estimation

**Hao-Ran Yang**[*]
Sun Yat-Sen University
Guangzhou, China
yanghr26@mail2.sysu.edu.cn

**Xiaohui Chen**[*]
Sun Yat-Sen University
Guangzhou, China
chenxh278@mail2.sysu.edu.cn

**Chuan-Xian Ren**[†]
Sun Yat-Sen University
Guangzhou, China
rchuanx@mail.sysu.edu.cn

## Abstract

Aiming to generalize the well-trained gaze estimation model to new target domains, Cross-domain Gaze Estimation (CDGE) is developed for real-world application scenarios. Existing CDGE methods typically extract the domain-invariant features to mitigate domain shift in feature space, which is proved insufficient by Generalized Label Shift (GLS) theory. In this paper, we introduce a novel GLS perspective to CDGE and modelize the cross-domain problem by label and conditional shift problem. A GLS correction framework is presented and a feasible realization is proposed, in which an importance reweighting strategy based on truncated Gaussian distribution is introduced to overcome the continuity challenges in label shift correction. To embed the reweighted source distribution to conditional invariant learning, we further derive a probability-aware estimation of conditional operator discrepancy. Extensive experiments on standard CDGE tasks with different backbone models validate the superior generalization capability across domain and applicability on various models of proposed method.

## 1 Introduction

Gaze estimation (GE) is crutial for understanding human attention in many areas, such as human-robot interaction [40], virtual reality [33] and medical analysis [24]. Recently, leveraging the capabilities of deep models, appearance-based gaze estimation [48, 25, 17] has attracted wide attention due to its low device requirements and end-to-end workflows. However, the great performance of deep models relies on the identical assumption of training and testing data distributions, which is hard to fulfill in real-world applications. When being applied across data domain, models' performance usually degrade dramatically due to the domain shift caused by the changes of subjects, environment, etc. Therefore, dealing with the cross-domain gaze estimation (CDGE) problem is crutial for expanding application prospects of gaze estimation.

Existing CDGE methods can be divided into two categories: Domain Generalization (DG) [5, 32, 43] and Unsupervised Domain Adaptation (UDA) [10, 1]. Technically, DG methods mainly focus on removing gaze-irrelevant factors from samples to obtain domain-invariant features that are generalizable to unseen target domains. And UDA methods typically align the feature distributions

---

[*]Equal contribution.
[†]Corresponding author.

so the predictor trained on them can be generalized to the specific target domain. Therefore, existing methods can be essentially summarized as the domain-invariant representation learning methods.

In this paper, we introduce a generalized label shift (GLS) perspective to CDGE problem. Formally, the GLS consists of distribution shifts in both the label and conditional distribution. We point out that in CDGE problem the difference of gaze range and concentration area results in label shift, while the conditional shift often arises from the difference of data collection environment. Integrating these two shifts, the CDGE problem is characterized as a GLS problem. According to GLS theory [31, 23], when the label shift exists, the invariant representation learning is insufficient to correct the domain shift. Therefore, in contrast to previous work, we introduce the GLS correction framework as a new paradigm to settle the CDGE problem. An intuitive illustration is provided in Fig. 1.

Existing GLS correction methods are primarily developed for classification problems, in which the label variables are discrete and finite. The label shift correction methods typically require iteration over all classes to estimate the class-wise distribution proportions, which is impractical for regression problems with continuous label variables, such as gaze variable. On the other hand, the conditional distribution alignment method proposed for regression problem such as Conditional Operator Discrepancy (COD), is operator-based method that entails the expectation operation on label variable, which is vulnerable to the label shift. Therefore, we propose a GLS correction method that is feasible for gaze estimation. Specifically, the label distribution is modeled as a bivariate truncated Gaussian distribution, based on which the continuous importance weight function is estimated to correct the label shift. Further, a probability-aware estimation of COD is derived, which enables the operator embedding of reweighted source distribution. Generally, our contributions are summarized as follows:

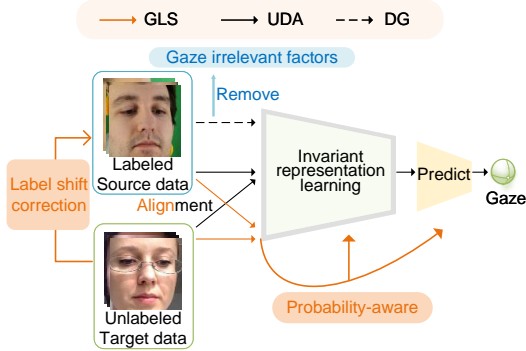

Figure 1: Illustration of GLS correction framework. The DG and UDA methods typically learn the invariant representation across domains. Differently, the proposed GLS correction framework consider both the label shift correction and the probability-aware invariant representation learning. Note that the color of the "Alignment" component is split into orange and black to indicate that both GLS and UDA perform distribution alignment.

- A novel GLS perspective is introduced and the CDGE problem is characterized as a GLS problem. The insufficiency of existing CDGE methods for successful cross-domain generalization is inferred from the GLS theory. Then a GLS correction framework is presented as a new paradigm to settle the CDGE problem.

- A GLS correction method feasible for CDGE problem is further developed to overcome the challenges arise from the continuity of gaze variable, in which a continuous importance reweighting strategy is proposed and a probability-aware estimation of COD is derived.

- Extensive validations are conducted on standard cross-domain tasks with different backbone models. The proposed GLS correction method achieve SOTA performance in comparison experiments, and significantly reduces the prediction error by 27.2%, 26.3%, 19.1% and 12.1% on four backbone models. The evaluation results highlight its superior generalization capability in cross-domain tasks and applicability across backbone models.

## 2 Preliminary

### 2.1 Problem Setup

Let $X$ and $Y$ be image and gaze variables defined on $\mathcal{X}$ and $\mathcal{Y}$, $\mathcal{D}^s = \{(x^s, y^s)\}$ and $\mathcal{D}^t = \{(x^t, y^t)\}$ be source and target domains with elements sampled from variables following different distributions, i.e., $(x^s, y^s) \sim (X^s, Y^s) \sim P^s_{XY}$ and $(x^t, y^t) \sim (X^t, Y^t) \sim P^t_{XY}$. For distribution $P$, the lowercase $p$ is the probability density function. Denote $g : X \to Z \in \mathcal{Z}$ as the feature transformation, $h : Z \to Y$ as the gaze predictor, then a learning model can be regarded as a tuple $(g, h)$. Let

$\ell(\cdot, \cdot) : \mathcal{Y} \times \mathcal{Y} \to \mathbb{R}_+$ be a loss function, then the CDGE problems aim to learn the model that minimizes the prediction error in target domain as

$$\underset{g,h}{\arg\min} \, \varepsilon^t(h \circ g) = \mathbb{E}_{P_{XY}^t} \left[ \ell(h(g(X)), Y) \right], \tag{1}$$

with only source samples and unlabeled target samples or only labeled source samples available.

## 2.2 Cross-domain Gaze Estimation (CDGE)

Existing CDGE methods can be roughly categorized as DG and UDA methods:

**DG methods.** Puregaze [5] and GazeCon [39] tackle the CDGE problems by gaze feature purification [3], which eliminate the gaze-irrelevant factors to obtain gaze-specific feature. GFAL [38] utilizes the gaze frontalization process and CUDA-GHR [12] adopts gaze redirection [44] as auxiliary learning task to improve the generalization capability of features. An attractive advantage of DG methods is that they don't need target domain samples and the gaze features can be generalized to unseen target domain. However, the improvement is usually limited due to the absence of knowledge from target domain.

**UDA methods.** Given some unlabeled samples from target domain, the UDA methods enhance the model performance in target domain by distribution alignment. DAGEN [10], PnP-GA [20] and PnP-GA+ [18] align the source and target distributions by different metrics while CRGA [34] and HFC [19] minimize the discrepancy in a contrastive learning paradigm. Recently, UnReGA [4], DUCA [46] propose to extract domain-invariant representations by reducing the uncertainty within features. Although these DG and UDA methods differ in technology, they essentially focus on domain-invariant representation learning.

## 2.3 Generalized Label Shift (GLS) Theory

A joint distribution $P_{XY}$ can be factorized as $P_{XY} = P_{X|Y} P_Y$. Then the discrepancy between $P_{XY}^s$ and $P_{XY}^t$ is naturally deduced to the following two shifts:

$$P_Y^s \neq P_Y^t, \qquad\qquad \text{(label shift)} \tag{2}$$

$$P_{X|Y}^s \neq P_{X|Y}^t. \qquad\qquad \text{(conditional shift)} \tag{3}$$

Most of the works concentrate on correcting one of them and assume the other shift has been corrected. Specifically, conditional shift correction methods [21, 22] aim to learn conditional invariant transformation $g : X \to Z$ such that $P_{Z|Y}^s = P_{Z|Y}^t$, while label shift correction methods [16, 9, 42] estimate reweighting strategy $\omega$ for source label distribution such that $P_Y^t = P_{Y\omega}^s \triangleq \omega P_Y^s$. By integrating the label and conditional shift, GLS problem is developed as a general setting on the joint distribution. Recent theoretical results [50, 31, 23] have verified the sufficiency and necessity of GLS correction for domain shift minimization:

- (**Necessity**) If the label shift exists, the invariant representation learning is insufficient to minimize the domain shift.
- (**Sufficiency**) The domain shift are sufficiently bounded by the label shift and conditional shift between domains.

However, existing GLS methods [45, 30, 29, 14] are developed for classification scenarios with finite classes. The typical class probability ratio estimation [35] and class-wise conditional alignment methods [27] are infeasible for regression problems with continuous and infinite label variable.

## 3 GLSGE: Perspective and Methodology

Here we provide a brief overview of the following sections. In Sec. 3.1, we introduce a novel GLS perspective to CDGE problem. In Sec. 3.2, we present the GLS correction framework and point out that existing DG and UDA methods can be viewed as partial realizations of this framework. In Sec. 3.3 and Sec. 3.4, we provide a feasible realization of the framework, in which a continuous importance reweighting strategy and a probability-aware estimation of COD are proposed to overcome the challenges arise from the continuity of gaze variable. In Sec. 3.5j, we summarize the proposed GLS correction method.

## 3.1 A GLS Perspective of CDGE problem

By the triangle inequality, the target prediction error can be easily factorized as

$$\underbrace{\varepsilon^t(h \circ g)}_{\text{target error}} \leq \underbrace{\varepsilon^s(h \circ g)}_{\text{source error}} + \underbrace{|\varepsilon^s(h \circ g) - \varepsilon^t(h \circ g)|}_{\text{domain shift}},$$

(4)

where the last term is often regarded as the domain shift. Therefore, solving Eq. (1) with only source domain samples and unlabeled target domain samples available is reduced to minimize the source prediction error $\varepsilon^s(h \circ g)$ and the domain shift $|\varepsilon^s(h \circ g) - \varepsilon^t(h \circ g)|$. As mentioned above, the domain shift is bounded by the GLS problem between domains. And we further point out that the CDGE problem can be indeed characterized as the GLS problem. An intuitive illustration of the characterization is shown in Fig. 2. The detailed analyses are as follows:

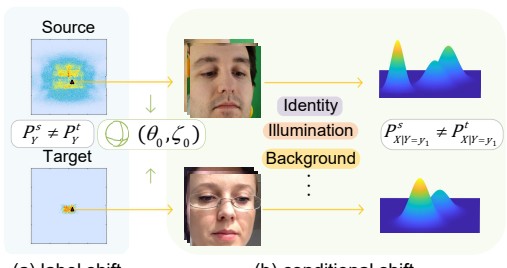

(a) label shift      (b) conditional shift

Figure 2: A GLS perspective of CDGE problem. (a) The label probability functions differ in functions support (colored area) and probability values (color degree), resulting the label shift between domains. (b) Conditional distributions of the same gaze label may differ due to factors like identity, background and illuminations.

**Label shift.** In CDGE problem, the source domain is usually larger than the target domain, in the meaning that the gaze distribution of source domain has a larger support. Meanwhile, in different application scenarios, people may focus their attention in different directions, leading to variety of gaze concentration area. So the probability of the same gaze may also differ. Generally, the observation above can be described as

$$supp\ p_Y^t \neq supp\ p_Y^s,$$
$$P_Y^t(y) \neq P_Y^s(y),\ \exists\ y \in supp\ p_Y^t \cap supp\ p_Y^s.$$

(5)

where $supp\ P_Y$ is the support of the gaze distribution. These formula induce the label shift Eq. (2).

**Conditional shift.** The difference in sample level is more intuitive. In different scenarios the appearance image may differ significantly due to the background, illuminations, etc, leading to the discrepancy of conditional distributions, i.e.,

$$P_{X|y}^s \neq P_{X|y}^t,\ \exists\ y \in supp\ P_Y^t \cap supp\ P_Y^s.$$

(6)

Summarizing Eq. (5) and Eq. (6), the CDGE problem can be modeled as a GLS problem.

*Remark* 3.1. From a probabilistic perspective, both factorizations of the joint distribution—i.e., $P(X,Y) = P(X|Y)P(Y)$ or $P(Y|X)P(X)$ —are mathematically valid. However, we adopt the factorization $P(X,Y) = P(X|Y)P(Y)$ because it provides a more intuitive and interpretable modeling perspective for the CDGE task. Further discussion can be found in Appendix B.

## 3.2 GLS correction framework for CDGE

Now we introduce the GLS correction framework for CDGE problem as follows:

- Estimating the importance weight function $\omega(y)$ and reweighted $P_{Y\omega}^s \triangleq \omega(y)P_Y^s$ by

$$\omega = \arg\min_{\omega} \mathcal{L}_{\text{lab}}(P_{Y\omega}^s, P_Y^t).$$

(7)

- Learning conditional invariant transformation $g$ with reweighted source distribution:

$$g = \arg\min_{g} \mathcal{L}_{\text{cond}}(P_{Z|Y\omega}^s, P_{Z|Y}^t).$$

(8)

- Learning gaze predictor $h$ on reweighted $P_{ZY\omega}^s$ by

$$h = \arg\min_{h} \mathcal{L}_{\text{src}}(h_\# P_Z^s, P_{Y\omega}^s),$$

(9)

where the $h_\# P_Z^s$ denotes the pushforward distribution.

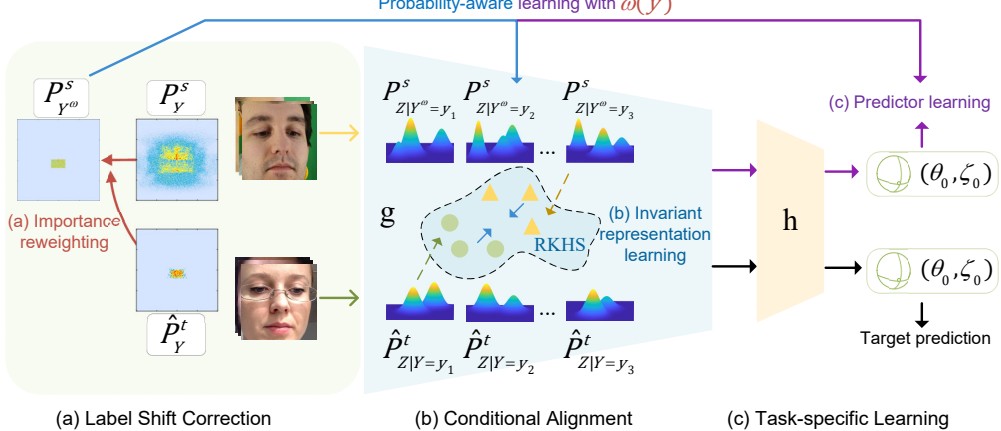

(a) Label Shift Correction    (b) Conditional Alignment    (c) Task-specific Learning

Figure 3: Overview of the proposed GLS correction method. (a) Reweighting source label distribution with bivariate Gaussian distribution estimated by pseudo target labels. The reweighted label distribution is utilized in both conditional alignment and task-specific training. (b) Conditional invariant representation learning. Two sets of conditional distribution are map to RKHS and the discrepancy is measured by the PCOD. (c) After label and conditional shift correction, i.e., the GLS correction, predictor trained on source domain can be generalized to target domain. Note that the text color reflects the component each learning objective is associated with.

The terms $\mathcal{L}_{\{\}}(\cdot, \cdot)$ denotes the learning Objectives in each step. By considering different steps and distance on distributions, **this framework is connected with existing CDGE methods**.

**DG methods** neglects the label shift since the target domains are unknown in the training stage, which means $\omega \equiv 1$. Intuitively, DG methods aim at a feature transformation $g$ that only extract task-specific factors from samples of any domain. In other words, the feature distributions of all domains are aligned to be the task-specific feature distribution. Formally, considering a distribution set $\mathbb{P}^t = \{P^i_{XY}\}_{i \in \mathcal{I}}$ in which all potential target distributions are involved, the generalization training process can be described as

$$\arg \min_{g,h} \mathcal{L}_{\mathrm{src}}(h_\# P^s_Z, P^s_Y) + \lambda \sup_{i \in \mathcal{I}} \mathcal{L}_{\mathrm{DG}}(P^s_Z, P^i_Z). \tag{10}$$

Intuitively, a feature transformation is well generalized to any unseen target domain only when the extracted feature contains all task-specific information and nothing else, which is extremely hard to obtain as the target distribution set $\mathbb{P}^t$ is unknown and unreachable.

**UDA methods** greatly improve the generalization capability with a few unlabeled target domain samples available. However, existing UDA methods in CDGE only consider the domain-invariant feature learning and neglect the label shift problem. The training process can be abstracted as:

$$\arg \min_{g,h} \mathcal{L}_{\mathrm{src}}(h_\# P^s_Z, P^s_Y) + \lambda \mathcal{L}_{\mathrm{UDA}}(P^s_Z, P^t_Z), \tag{11}$$

where the $\mathcal{L}_{\mathrm{UDA}}$ refers to marginal alignment loss or conditional alignment loss. The ideal $(g, h)$ in UDA method is easier to attain than in DG as there is some unlabeled samples accessible.

In summary, previous works are essentially connected with the invariant representation learning and omit the label shift correction. According to the GLS theory, such methods are insufficient for successful cross domain learning when the label shift exists. Therefore, beyond previous methods, we propose to tackle CDGE problem based on the GLS correction framework. In the following sections, we provide a feasible realization of proposed framework. Specifically, we (1) modelize the gaze distribution as bivariate truncated Gaussian distribution for label shift correction and (2) derive an probability-aware estimation of conditional distribution discrepancy.

### 3.3 Label Probability Distribution Estimation

Due to the continuity of gaze variable, it is impractical to correct label shift by existing classification-based methods that iterate over all classes to compute the class-wise distribution proportions. In fact,

even though the number of distinct label values is finite in training stage, the gaze variable still admits the continuous property, which means the discrete clustering is still infeasible. Therefore, we propose a continuous importance reweighting method that is feasible for CDGE problem.

Note that unlike in general regression tasks, the gaze variable has a specific limited range. In fact, the range of gaze can be considered as a rectangular region (or a spherical region, depending on the choice of coordinate system). From a mathematical perspective, the gaze distribution has a compact support. Besides, in real-world scenarios, people's gaze is primarily concentrated in a certain area, with probability becoming lower towards the edges of the support. With these priors, we modelize the label distribution as a bivariate truncated Gaussian distribution with probability density function:

$$
\begin{aligned}
p_Y(y) &= f_{\mathrm{TGau}}(y; \mu, \Sigma, a, b) \\
&= \frac{\mathbf{1}_{a \times b}(y)\, f_{\mathrm{Gau}}(y; \mu, \Sigma)}{F_{\mathrm{Gau}}(v_{22}; \mu, \Sigma) - F_{\mathrm{Gau}}(v_{12}; \mu, \Sigma) - F_{\mathrm{Gau}}(v_{21}; \mu, \Sigma) + F_{\mathrm{Gau}}(v_{11}; \mu, \Sigma)},
\end{aligned}
\tag{12}
$$

where $y = (y_1, y_2)^T$ is the gaze value, $\mathbf{1}_{a \times b}(y)$ is the characteristic function. Let $v_{11} = (a_1, b_1)^T, v_{12} = (a_1, b_2)^T, v_{21} = (a_2, b_1)^T, v_{22} = (a_2, b_2)^T$ denote the vertices of the rectangular range $(a_1, a_2) \times (b_1, b_2)$, $a := (a_1, a_2)$ and $b := (b_1, b_2)$ denote the intervals, $f_{\mathrm{Gau}}(y; \mu, \Sigma)$ and $F_{\mathrm{Gau}}(y; \mu, \Sigma)$ denote the probability and cumulative density functions of bivariate Gaussian distribution with mean $\mu$ and covariance $\Sigma$. By estimating the statistics $(\mu, \Sigma)$ with target pseudo label $\hat{y}^t = h(g(x^t))$, the importance weight function takes the form

$$
\omega(y) = \frac{f_{\mathrm{TGau}}(y; \hat{\mu}^t, \hat{\sigma}^t, a, b)}{p_Y^s},
\tag{13}
$$

Then for any $y \in supp\, p_Y^s$, the reweighted probability can be directly approximated by

$$
p_{Y^\omega}^s(y) = \omega(y) p_Y^s(y) = f_{\mathrm{TGau}}(y; \hat{\mu}^t, \hat{\sigma}^t, a, b),
\tag{14}
$$

where the interval vectors a and b are determined by the confidence area of Gaussian distribution.

Note that **the truncated Gaussian distribution is only an alternative to model the gaze distribution**. Other continuous distributions with compact support, such as exponential family distributions, uniform distributions, or even shallow neural networks, can also be utilized. The choice of $p_Y$ can be adapted to the characteristics of the specific task or dataset, e.g., in driver gaze datasets where gaze points tend to concentrate around a few specific directions like forward, mirrors, a truncated Gaussian mixture model may provide a more accurate fit.

## 3.4 Probability-Aware Conditional Alignment

Most of the existing conditional invariant learning methods are developed for classification problem and are trapped in similar dilemma with label shift correction methods in regression problem, i.e., the class-wise distribution alignment induces infinite matching problems for $P_{X|y}$ on every possible $y$ due to the continuity of label variable. To overcome this inevitable obstacle, latest work [41] propose a Conditional Operator Discrepancy (COD) that admits the metric property on conditional distributions via the kernel embedding theory to align all $P_{X|y}$ as a whole $P_{X|Y}$:

$$
d_{\mathrm{COD}}^2(P_{Z|Y}^s, P_{Z|Y}^t) = \|\mathcal{U}_{Z|Y}^s - \mathcal{U}_{Z|Y}^t\|_{\mathcal{H}_\mathcal{K}}^2 + \mathrm{tr}(\mathcal{C}_{ZZ|Y}^{ss} + \mathcal{C}_{ZZ|Y}^{tt} - 2\mathcal{C}_{ZZ|Y}^{st}),
\tag{15}
$$

where $\mathcal{U}_{Z|Y}$ and $\mathcal{C}_{ZZ|Y}$ are conditional operators in Reproducing Kernel Hilbert Space (RKHS). Note that COD embeds the whole label distribution $P_Y$ rather than embeds $P_Y(y)$ one by one to the RKHS. And it contains the expectation operation on the label variable. Thus, the empirical estimation provided in [41] is not capable of reweighted label distribution embedding. For example, the empirical estimation of $\mathbb{E}_{P_Y}[r(Y)]$ for any function $r(y)$ is $\sum_y \hat{p}_Y(y) r(y) = \frac{1}{n} \sum_{i \in \mathcal{I}} r(y_i)$, regardless of the prior of distribution $P_Y$. In contrast, a probability-aware estimation with importance weight function $\omega(y)$ is

$$
\sum_y \hat{p}_{Y^\omega}(y) r(y) = \sum_{i \in \mathcal{I}} \omega(y_i) \hat{p}(y_i) r(y_i) = \frac{1}{n} \sum_{i \in \mathcal{I}} \omega(y_i) r(y_i).
\tag{16}
$$

Such change in estimation differs the distribution embedding. As we aim to align $P_{Z|Y}^t$ with the reweighted $P_{Z|Y^\omega}^s$ rather than $P_{Z|Y}^s$, it is necessary to derive a probability-aware estimation of COD,

denoted by PCOD. We carefully complete the derivation, which involves extensive probability and matrix analysis. Here we directly present the results of the derivation and provide the details of the derivation process in Appendix C.

Let $\{(\mathbf{x}_i^s, \mathbf{y}_i^s)\}_{i=1}^n$ and $\{(\mathbf{x}_i^t, \mathbf{y}_i^t)\}_{i=1}^n$ be two sets of samples drawn i.i.d. from source and target domain, $\mathbf{z}_i^s = g(\mathbf{x}_i^s)$, $\mathbf{z}_i^t = g(\mathbf{x}_i^t)$ be the feature extracted by $g$, $k_{\mathcal{Z}} : \mathcal{Z} \times \mathcal{Z} \to \mathbb{R}$, $k_{\mathcal{Y}} : \mathcal{Y} \times \mathcal{Y} \to \mathbb{R}$ be the kernel functions on $\mathcal{Z}$ and $\mathcal{Y}$. The kernel matrix is computed as $(\mathbf{K}_Z^{ss})_{ij} = k_{\mathcal{Z}}(\mathbf{z}_i^s, \mathbf{z}_j^s)$, and so as $\mathbf{K}_Z^{tt}$, $\mathbf{K}_Z^{ts}$, $\mathbf{K}_Y^{ss}$, $\mathbf{K}_Y^{tt}$ and $\mathbf{K}_Y^{ts}$. Let $\mathbf{q} = (q_1, \ldots, q_n)^T$ be the discretization vector of reweighted source label distribution $p_{Y^\omega}$, n be the number of distinct source sample labels. Then

$$
\begin{aligned}
&\widehat{d}_{\mathrm{PCOD}}^2(P_{Z|Y^\omega}^s, P_{Z|Y}^t)\\
=&\mathrm{tr}(\mathbf{K}_Z^{ss}\mathbf{Q}(\mathbf{K}_Y^{ss}\mathbf{Q} + \varepsilon\mathbf{I})^{-1}\mathbf{K}_Y^{ss}(\mathbf{Q}\mathbf{K}_Y^{ss} + \varepsilon\mathbf{I})^{-1}\mathbf{Q})\\
&+ \mathrm{tr}(\mathbf{K}_Z^{tt}(\mathbf{K}_Y^{tt} + \varepsilon n\mathbf{I})^{-1}\mathbf{K}_Y^{tt}(\mathbf{K}_Y^{tt} + \varepsilon n\mathbf{I})^{-1})\\
&- 2\mathrm{tr}(\mathbf{K}_Z^{ts}\mathbf{Q}(\mathbf{K}_Y^{ss}\mathbf{Q} + \varepsilon\mathbf{I})^{-1}\mathbf{K}_Y^{st}(\mathbf{K}_Y^{tt} + \varepsilon n\mathbf{I})^{-1}) + \varepsilon\mathrm{tr}\left[\mathbf{G}_{Z^\omega}^s(\mathbf{G}_{Y^\omega}^s + \varepsilon\mathbf{I}_n)^{-1}\right]\\
&+ \varepsilon\mathrm{tr}\left[\mathbf{G}_Z^t(\mathbf{G}_Y^t + \varepsilon n\mathbf{I}_n)^{-1}\right] - \frac{2}{\sqrt{n}}\left\|\mathbf{M}^T\mathbf{K}_Z^{ts}\mathbf{M}^\omega\right\|_*,
\end{aligned}
\tag{17}
$$

where the $\|\cdot\|_*$ is the nuclear norm, $\mathbf{M}$ and $\mathbf{M}^\omega$ are defined by $\mathbf{M}\mathbf{M}^T = \mathbf{H}_n\varepsilon(\mathbf{G}_Y^t + \varepsilon\mathbf{I}_n)^{-1}$ and $\mathbf{M}^\omega\mathbf{M}^{\omega T} = \mathbf{B}\varepsilon(\mathbf{G}_{Y^\omega}^t + \varepsilon\mathbf{I}_n)^{-1}\mathbf{B}^T$. Other notations used in the formula are summarized in Tab. 1.

Note that in practice the target label $\mathbf{y}_i^t$ is replaced by target pseudo label. When the domain shift is significant, it is difficult for the predictor to produce high-quality pseudo labels at the begining of training process, which may trap the model into sub-optimal solutions. For this reason, we add a marginal alignment term to improve the reliability of pseudo labels as done in [41]. Then the probability-aware conditional alignment loss is formulated as

| | |
|---|---|
| $\mathbf{1}_n = (1, \ldots, 1)_n^T$ | $\mathbf{I}_n = diag(\mathbf{1}_n)$ |
| $\mathbf{H}_n = \mathbf{I}_n - \frac{1}{n}\mathbf{1}_n\mathbf{1}_n^T$ | $\mathbf{G} = \mathbf{H}_n\mathbf{K}\mathbf{H}_n$ |
| $\mathbf{Q} = diag(\mathbf{q})$ | $\mathbf{H}_q = \mathbf{Q} - \mathbf{q}\mathbf{q}^T = \mathbf{B}\mathbf{B}^T$ |
| $\mathbf{B} = (\sqrt{\mathbf{Q}} - \mathbf{q}\sqrt{\mathbf{q}}^T)$ | $\mathbf{G}_{Y^\omega} = \mathbf{B}^T\mathbf{K}_Y\mathbf{B}$ |

Table 1: Part of notations used in PCOD

$$
\mathcal{L}_{\mathrm{cond}} = \widehat{d}_{\mathrm{PCOD}}(P_{Z|Y^\omega}^s, P_{Z|Y}^t) + \widehat{d}_{\mathrm{marg}}(P_Z^s, P_Z^t),
\tag{18}
$$

where the $\widehat{d}_{\mathrm{marg}}$ can be any marginal distribution discrepancy metric. In this paper we adopt the DAGE-GRAM [26] that is proposed for regression problem.

As the PCOD is derived for the continuity and label shift of gaze variable, it is applicable to other tasks with similar distributional properties, providing a more general-purpose solution within the GLS correction framework. Of course, beyond distributional aspects, gaze estimation has other unique properties, such as rotational equivariance. Integrating these domain-specific priors into the GLS correction method is an interesting direction for future work.

### 3.5 GLS Correction Model for CDGE

Combining Sec. 3.3 and Sec. 3.4, we obtain a practical realization of the GLS correction framework:

$$
p_{Y^w}^s(y) = f_{TGau}(y; \hat{\mu}^t, \hat{\sigma}^t, a, b), \quad (g, h) = \arg\min_{g,h} \mathcal{L}_{src}^\omega + \lambda\mathcal{L}_{\mathrm{cond}}
\tag{19}
$$

where $\lambda$ is the trade-off parameter. The task-specific loss $\mathcal{L}_{src}$ is the reweighted $\mathcal{L}_1$ loss:

$$
\mathcal{L}_{src}^\omega = \sum_{i=1}^{n^s} p_{Y^\omega}^s(\mathbf{y}_i^s)\|h(g(\mathbf{x}_i^s)) - \mathbf{y}_i^s\|_1.
\tag{20}
$$

An intuitive illustration is shown in Fig. 3. For convenience, We shortly name our works as GLSGE. A form of algorithm and computational efficiency analysis are provided in Appendix A.2.

Note that the introduced GLS correction framework is not limited to gaze estimation and can be potentially extended to a broad range of tasks with similar distributional properties such as pose estimation. We would also like to emphasize that our contribution is not only in the formulation of a general framework, but also in the design of a feasible and effective implementation for CDGE. We believe this step is crucial when adapting the general GLS framework to different tasks, and it often requires task-specific insights and custom solutions.

## 4 Experiment

**Setups.** We conduct experiments on four standard CDGE datasets: ETH-XGaze ($\mathcal{D}_E$) [47],Gaze360 ($\mathcal{D}_G$) [13],MPIIFaceGaze ($\mathcal{D}_M$) [49] and EyeDiap ($\mathcal{D}_D$) [8]. The face image data are normalized according to the preparation process summarized in [7]. Details about these datasets are provided in Appendix A.1. Following previous works, we adopt $\mathcal{D}_E, \mathcal{D}_G$ as source domains and $\mathcal{D}_M, \mathcal{D}_D$ as target domains, as the former two domains have wider gaze distributions than the latter. Then four CDGE tasks are established: $\mathcal{D}_E \to \mathcal{D}_M, \mathcal{D}_E \to \mathcal{D}_D$, $\mathcal{D}_G \to \mathcal{D}_M$ and $\mathcal{D}_G \to \mathcal{D}_D$. During cross-domain learning, we use 10% of the unlabeled target domain images for training and another 10% for validation, with the remaining 80% used for testing. It means that 4500 images in $\mathcal{D}_M$ and 1667 images in

| Method | $\mathcal{D}_E \to \mathcal{D}_M$ | $\mathcal{D}_E \to \mathcal{D}_D$ | $\mathcal{D}_G \to \mathcal{D}_M$ | $\mathcal{D}_G \to \mathcal{D}_D$ | Avg |
|---|---|---|---|---|---|
| ResNet-18 | 8.05 | 9.03 | 7.41 | 8.83 | 8.33 |
| PureGaze | 9.14 | 8.37 | 9.28 | 9.32 | 9.03 |
| GazeCon | 6.50 | 7.44 | 7.55 | 9.03 | 7.63 |
| GFAL | 5.72 | 6.97 | 7.18 | 7.38 | 6.81 |
| AGG | 7.10 | 7.07 | 7.87 | 7.93 | 7.49 |
| FSCI | 5.79 | 6.96 | 7.06 | 7.99 | 6.95 |
| CGaG | 6.47 | 7.03 | 7.50 | 8.67 | 7.42 |
| DAGEN | 5.73 | 6.77 | 7.38 | 8.00 | 6.97 |
| PnP-GA | 5.53 | 5.87 | 6.18 | 7.92 | 6.38 |
| GSA-Gaze | 6.45 | 7.25 | 6.37 | 8.95 | 7.26 |
| HFC | 5.35 | 6.24 | 7.18 | 8.61 | 6.84 |
| DCUA | 7.31 | 5.95 | 5.59 | **6.4** | 6.31 |
| PnP-GA+ | 5.34 | **5.73** | 6.10 | 7.62 | 6.20 |
| GLSGE | **5.31** ±0.02 | 6.21 ±0.04 | **5.43** ±0.03 | 7.30 ±0.02 | **6.06** ±0.03 |

Table 2: Comparision with ResNet-18 as backbone model.

$\mathcal{D}_D$ are used for training in each task. Further discussion about the influence of target training data size is provided in Appendix A.3, while implementation details are provided in Appendix A.4. Each evaluation experiment is repeated five times and the average results are reported.

### 4.1 Comparison with SOTA methods

**Comparison Methods.** We compare GLSGE with (1) DG methods: PureGaze [5], GFAL [38], AGG [2], FSCI [15], CGaG [37] and (2) UDA methods: DAGEN [10], PnP-GA [20], GSA-Gaze [11], HFC [19], DCUA [46], PnP-GA+ [18]. Following previous works, the evaluation are based on ResNet-18 and ResNet-50. Angular error in degrees are reported as evaluation metric for all tasks.

**Results with ResNet-18.** In Tab. 2, we compare GLSGE with SOTA methods that use ResNet-18 as backbone model. Note that the enhancement approaches such as data augmentation that are widely used for improving performance are not applied in GLSGE. Nevertheless, GLSGE still exhibits superior performance among SOTA methods in average results and attains the best performance in $\mathcal{D}_E \to \mathcal{D}_M$ and $\mathcal{D}_G \to \mathcal{D}_M$ tasks.

**Results with ResNet-50.** As shown in Tab. 3, GLSGE achieves comparable performance in most of tasks and outperforms existing methods in average result. In particular, GLSGE largely reduces the prediction error in task $\mathcal{D}_G \to \mathcal{D}_M$ (from 5.61 to 5.27 compared with the second best) and

| Method | $\mathcal{D}_E \to \mathcal{D}_M$ | $\mathcal{D}_E \to \mathcal{D}_D$ | $\mathcal{D}_G \to \mathcal{D}_M$ | $\mathcal{D}_G \to \mathcal{D}_D$ | Avg |
|---|---|---|---|---|---|
| ResNet-50 | 8.03 | 8.06 | 7.75 | 8.79 | 8.16 |
| PureGaze | 7.08 | 7.48 | 7.62 | 7.70 | 7.47 |
| GSA-Gaze | 7.62 | 8.14 | 9.83 | 10.02 | 8.90 |
| GazeCon | 8.35 | 8.8 | 8.24 | 8.83 | 8.56 |
| AGG | 5.91 | 6.75 | 9.20 | 11.36 | 8.31 |
| FSCI | **5.47** | 6.68 | 6.19 | 10.20 | 7.14 |
| PnP-GA | 6.58 | 6.79 | 5.70 | 7.14 | 6.55 |
| HFC | 7.68 | 9.54 | 8.86 | 7.66 | 8.44 |
| DCUA | 7.58 | **5.65** | 5.61 | **6.11** | 6.24 |
| PnP-GA+ | 6.49 | 6.61 | 5.64 | 7.09 | 6.46 |
| GLSGE | 5.54 ±0.03 | 6.10 ±0.04 | **5.27** ±0.02 | 7.14 ±0.03 | **6.01** ±0.03 |

Table 3: Comparision with ResNet-50 as backbone model.

achieves the second best performance in tasks $\mathcal{D}_E \to \mathcal{D}_M$ and $\mathcal{D}_E \to \mathcal{D}_D$. Such evaluation results firmly verify the effectiveness of the proposed method.

| Method | $\mathcal{D}_E \to \mathcal{D}_M$ | $\mathcal{D}_E \to \mathcal{D}_D$ | $\mathcal{D}_G \to \mathcal{D}_M$ | $\mathcal{D}_G \to \mathcal{D}_D$ | Avg |
|---|---|---|---|---|---|
| Res-18 | 8.05 | 9.03 | 7.41 | 8.83 | 8.33 |
| Res-18 + GLSGE | 5.31 ▼ 34.0% | 6.21 ▼ 31.2% | 5.43 ▼ 26.7% | 7.30 ▼ 17.3% | 6.06 ▼ 27.2% |
| Res-50 | 8.03 | 8.06 | 7.75 | 8.79 | 8.16 |
| Res-50 + GLSGE | 5.54 ▼ 31.0% | 6.10 ▼ 24.3% | 5.27 ▼ 32.0% | 7.14 ▼ 18.8% | 6.01 ▼ 26.3% |
| GazeTR | 8.69 | 10.94 | 7.11 | 9.20 | 8.99 |
| GazeTR + GLSGE | 5.90 ▼ 32.1% | 9.07 ▼ 17.1% | 5.36 ▼ 24.6% | 8.75 ▼ 4.9% | 7.27 ▼ 19.1% |
| FSCI | 5.79 | 6.96 | 7.06 | 7.99 | 6.95 |
| FSCI + GLSGE | 5.38 ▼ 7.1% | 6.19 ▼ 11.1% | 6.08 ▼ 13.9% | 6.80 ▼ 14.9% | 6.11 ▼ 12.1% |

Table 4: Evaluation results of plugging GLSGE to different models and SOTA method. Angular error in degrees and error reduction percentages with GLSGE plugged are reported.

## 4.2 Applicability Analysis

As GLSGE is a general framework for CDGE problem, we futher investigate its applicability across different models. Apart from ResNet-18 and ResNet-50, we plug GLSGE into (1) GazeTR that employs vision transformer (ViT) as backbone and (2) FSCI, the newly proposed SOTA DG method. As shown in Tab. 4, for the most widely used **ResNet models**, GLSGE significantly boosts the performance by over 25% in average results. In particular, GLSGE reduces the prediction error by 34.0% and 31.2% in tasks $\mathcal{D}_E \to \mathcal{D}_M$ and $\mathcal{D}_E \to \mathcal{D}_D$ compared with the ResNet-18 baseline. For the ViT-based model **GazeTR**, GLSGE largely reduces the prediction error by 32.1% in task $\mathcal{D}_E \to \mathcal{D}_M$ and 19.1% in average. Moreover, although **FSCI** is already the SOTA DG method, GLSGE further reduces the prediction error by 12.1% in average. Note that the prediction error of FSCI + GLSGE is larger than ResNet-18 + GLSGE. The loss of task-specific factors during the DG training process might be the reason. Above all, the great improvement across models further supports the broad applicability of GLSGE and its potential value to the area as a plug-play method.

## 4.3 Ablation Analysis

The ablation experiment is conducted on tasks $\mathcal{D}_E \to \mathcal{D}_D$ and $\mathcal{D}_G \to \mathcal{D}_M$ with ResNet-18 as backbone model. The major components of proposed method are evaluated, i.e., the label shift correction and the conditional distribution alignment. The evaluation results are shown in Tab. 5. (1) From the $2^{nd}$-$4^{th}$ rows, it can be observed that both label shift correction and conditional alignment largely reduce the prediction error. In particular, the PCOD is more effective than the original COD method, while both of them are negatively affected by the label shift problem. (2) As

| Objectives | $\mathcal{D}_E \to \mathcal{D}_D$ | $\mathcal{D}_G \to \mathcal{D}_M$ |
|---|---|---|
| $\mathcal{L}_{src}$ | 9.03 | 7.41 |
| $\mathcal{L}_{src}^w$ | 6.70 | 6.37 |
| $\mathcal{L}_{src} + d_{\text{COD}}$ | 7.42 | 6.58 |
| $\mathcal{L}_{src} + d_{\text{PCOD}}$ | 7.15 | 6.44 |
| $\mathcal{L}_{src}^w + d_{\text{COD}}$ | 6.32 | 5.76 |
| $\mathcal{L}_{src}^w + d_{\text{PCOD}}$ | **6.21** | **5.43** |

Table 5: Ablation study.

shown in the $5^{th}$ and $6^{th}$ rows, the combination of label and contional shift correction, i.e. the GLS correction, gains prominent improvement in both tasks, which exhibits the mutually beneficial relation between the two modules. And the comparison between the 5th and 6th rows further verifies the reasonability of the probability-aware estimation for conditional distribution discrepancy.

## 4.4 Visualization Analysis of GLS Correction

To gain insights into how GLSGE improves model's generalizability in target domain, a visualization analysis on $\mathcal{D}_E \to \mathcal{D}_M$ is provided in Fig. 4. **(a)-(c):** Before GLS correction, the source label distribution has a much wider range than that of target distribution and the feature distributions differ severly, which implies the significant label and conditional shift between domains. The prediction error is large when the predictor trained in $\mathcal{D}_E$ is directly deployed in $\mathcal{D}_M$. **(d)-(f):** After GLS correction, the reweighted source label distribution overlaps with target label distribution to a large extent, while the conditional distributions are aligned and the features are matched according to gaze values. The changes of color shows a clear gradient of features so the labeling rules are similar for both domains. As shown in **(f)**, the prediction distribution is brought much closer to the ground-truth label distribution by GLS correction.

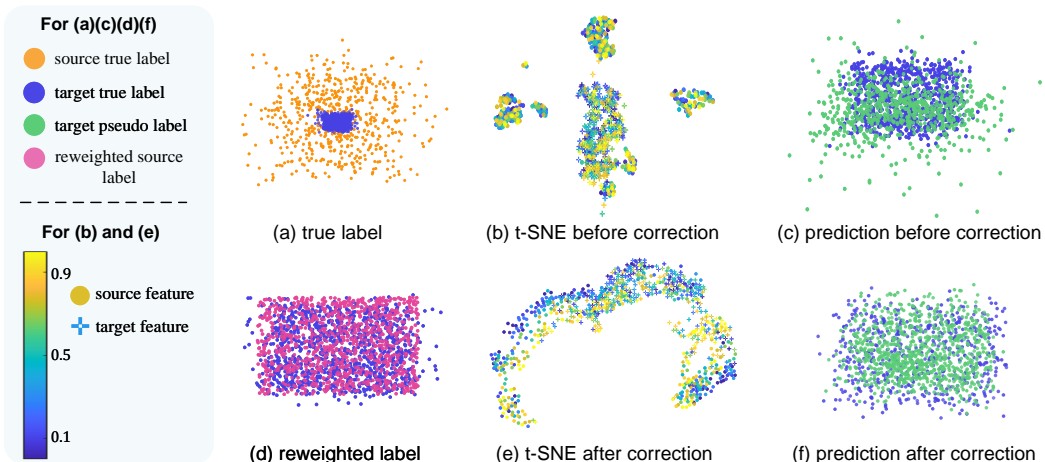

Figure 4: Visualization of GLS correction process. For scatter plot (a)(c)(d)(f), label variables are all denoted by '●' regardless of value and are distinguished from other domains by colors. In contrast, for t-SNE figure (b) and (e), label values are denoted by color gradients and source domain features are denoted by '●' while target features are denoted by '+'

## 4.5 Hyper-parameter

The sensitivity of hyper-parameters in Eq. (19) is analyzed on $\mathcal{D}_E \rightarrow \mathcal{D}_M$. The variation of confidence alters the truncated area in Eq. 12 while the $\lambda$ decides the participation of PCOD in the training process. The results shown in Fig. 5 validate the robustness of proposed GLSGE method. With a standard deviation of only 0.05, the deviation is slight in a wide range of parameters variation. Besides, the GLSGE is stable in random experiments, which also demonstrates the reliability of GLS correction.

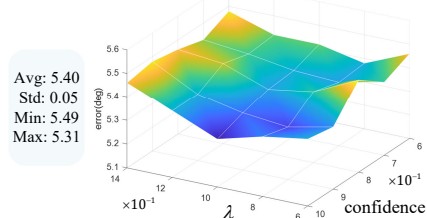

Figure 5: Prediction error under different settings of hyper-parameters.

## 5 Conclusion

In this work, we introduce a novel GLS perspective to CDGE and point out the label shift and conditional shift phenomena in cross-domain learning to characterize CDGE as a GLS problem. A GLS correction framework is then presented as a new paradigm for solving the CDGE problem due to the insufficiency of existing CDGE methods for successful cross-domain learning that are inferred from the GLS theory. To overcome the challenges arise from the continuity of gaze variable, we introduce the truncated Gaussian distribution for label importance reweighting and derive a probability-aware estimation of COD for conditional invariant learning. Numerical evaluation experiments on standard CDGE tasks with different backbone models validate the superior generalizability across domains and applicability on various models of proposed method.

**Limitations and future work. (1)** As domain shift correction is the core of GLSGE, target domain samples are required in training stage. While this enhances the model's generalization capability, its applicability to other unseen domains is limited. **(2)** The adopted conditional alignment method involves the computation of kernel matrices, which is computationally intensive. To address this issue, previous works such as random features can be employed to effectively reduce the computational load. More fundamentally, the proposed method is a feasible implementation of the GLSGE framework. Therefore, more efficient conditional distribution metrics suitable for continuous variables can also be applied within this framework, which will be a worthwhile direction for future exploration. **(3)** The provided label distribution correction method is a general but simple implementation. In real-world application, the label distribution should be estimated based on the prior knowledge of the problem. For instance, in the task of driver's gaze estimation, the label distribution can be modeled as a Gaussian mixture model. In summary, the effectiveness and efficiency of the proposed general GLSGE framework can be further enhanced by problem-specific prior in application scenarios.

## Acknowledgements

This work is supported in part by National Key R&D Program of China (2024YFA1011900), National Natural Science Foundation of China (Grant No. 62376291), Guangdong Basic and Applied Basic Research Foundation (2023B1515020004), Science and Technology Program of Guangzhou (2024A04J6413), and the Fundamental Research Funds for the Central Universities, Sun Yat-sen University (24xkjc013).

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

## Notations

| Notation | Description |
|---|---|
| $\mathcal{D}_s, \mathcal{D}_t$ | Source and target domain. |
| $\mathcal{X}, \mathcal{Y}, \mathcal{Z}$ | Input, output and feature space. |
| $X, Y, Z$ | Image, gaze and feature variables. |
| $x, y, z$ | Image, gaze and feature samples. |
| $\{(\mathbf{x}_i^s, \mathbf{y}_i^s, \mathbf{z}_i^s)\}_{i=1}^n$ | Vectorized source images, gazes and features samples. |
| $\{(\mathbf{x}_i^t, \mathbf{y}_i^t, \mathbf{z}_i^t)\}_{i=1}^n$ | Vectorized target images, gazes and features samples. |
| $n$ | Number of samples in source/target domain. |
| $(g, h)$ | Feature extractor and gaze estimator. |
| $h_\#$ | Push-forward operator of $h$. |
| $P_{X,Y}^s(x,y), P_{X,Y}^t(x,y)$ | Joint probability of $(X, Y)$ in source and target domain. |
| $P_X^s(x), P_X^t(x)$ | Marginal probability of $X$ in source and target domain. |
| $P_Y^s(y), P_Y^t(y)$ | Marginal probability of $Y$ in source and target domain. |
| $P_{X|Y}^s(x|y), P_{X|Y}^t(x|y)$ | Conditional probability of $X$ given $Y$ in source and target domain. |
| $P_{Z|Y}^s(z|y), P_{Z|Y}^t(z|y)$ | Conditional probability of $Z$ given $Y$ in source and target domain. |
| $P_{Y|X}^s(y|x), P_{Y|X}^t(y|x)$ | Conditional probability of $Y$ given $X$ in source and target domain. |
| $\mathbb{P}$ | Probability measure space. |
| $\ell(\cdot, \cdot)$ | Loss function. |
| $\varepsilon^s(), \varepsilon^t()$ | Source and target domain prediction error. |
| $\omega(y)$ | Label importance weighting function. |
| $\mathcal{L}$ | Learning objective. |
| $f_{\text{TGau}}$ | Probability density function of truncated Gaussian distribution. |
| $f_{\text{Gau}}$ | Probability density function of Gaussian distribution. |
| $F_{\text{Gau}}$ | Cumulative density function of Gaussian distribution. |
| $\hat{y}^t$ | Pseudo label of target gaze. |
| $\mu, \Sigma$ | Mean and covariance of Gaussian distribution. |
| $\hat{\mu}^t, \hat{\Sigma}^t$ | Estimated mean and covariance of target pseudo label. |
| $a = (a_1, a_2), b = (b_1, b_2)$ | Interval vectors of truncated Gaussian distribution. |
| $v_{11}, v_{12}, v_{21}, v_{22}$ | Vertices of the rectangular range $(a_1, a_2) \times (b_1, b_2)$. |
| $\mathcal{U}_{Z|Y}^s, \mathcal{U}_{Z|Y}^t$ | Conditional mean embedding of $P_{Z|Y}^s$ and $P_{Z|Y}^t$. |
| $\mathcal{C}_{ZZ|Y}^{ss}, \mathcal{C}_{ZZ|Y}^{tt}$ | Conditional covariance operators of $P_{Z|Y}^s$ and $P_{Z|Y}^t$. |
| $\mathcal{C}_{ZZ|Y}^{ts}$ | Cross conditional covariance operator between $P_{Z|Y}^t$ and $P_{Z|Y}^s$. |
| $\varepsilon$ | Regularization parameter. |
| $\lambda$ | Trade-off parameter. |
| $k_{\mathcal{Z}}, k_{\mathcal{Y}}$ | Kernel functions on $\mathcal{Z}$ and $\mathcal{Y}$. |
| $\mathbf{K}_Z^{ss}, \mathbf{K}_Z^{tt}, \mathbf{K}_Z^{ts}$ | Kernel matrices on $\mathcal{Z}$ for source-source, target-target and target-source samples. |
| $\mathbf{K}_Y^{ss}, \mathbf{K}_Y^{tt}, \mathbf{K}_Y^{ts}$ | Kernel matrices on $\mathcal{Y}$ for source-source, target-target and target-source samples. |
| $\mathbf{1}_n, \mathbf{I}_n$ | All-ones vector and identity matrix of size $n$. |
| $\mathbf{H}_n$ | Centering matrix of size $n$. |
| $\mathbf{G}$ | Centered kernel matrix. |
| $\mathbf{Q}$ | Diagonal matrix of discretization vector $\mathbf{q}$. |
| $\mathbf{H}_q$ | Centering matrix of $\mathbf{q}$. |
| $\mathbf{B}$ | Matrix defined by $\mathbf{H}_q = \mathbf{B}\mathbf{B}^T$. |
| $\mathbf{G}_{Y\omega}$ | Centered reweighted kernel matrix on $\mathcal{Y}$. |

Table 6: Notations used in this paper.

## A  Experimental details

### A.1  Datasets

**ETH-XGaze** ($\mathcal{D}_E$) **[47]** is collected under varying lighting conditions with custom hardware. It contains over one million high-resolution images of different gaze directions under extreme head poses. Following the original paper, we use 756,540 images from 80 participants as the training set.

**Gaze360** ($\mathcal{D}_G$) **[13]** is collected in both indoor and outdoor environments with a 360° camera. It contains data from 238 participants. We use only 84,902 images of frontal faces as the training set.

**MPIIFaceGaze** ($\mathcal{D}_M$) **[49]** is collected during daily usage of laptops, including data from 15 participants. Following the standard evaluation protocol, 3,000 images from each subject are used for evaluation.

**EyeDiap** ($\mathcal{D}_D$) **[8]** is collected in a laboratory environment with a screen and a floating ball as gaze targets, which includes multiple video clips from 16 participants. We use 16,674 images from the screen target sessions of 14 participants for evaluation.

### A.2 Algorithm

---

**Algorithm 1:** Optimization of GLSGE

---

1    **Input:** Source data $\{(\mathbf{x}_i^s, \mathbf{y}_i^s)\}_{i=1}^n$ and unlabeled target data $\{\mathbf{x}_i^t\}_{i=1}^n$, source pretrained model $(g, h)$;

2    **for** $N_1$ *steps* **do**

3       Estimate the reweighted label probability function $p_{Y^\omega}^s(y)$ by Eq. (14) ;

4       **for** $N_2$ *epoches* **do**

5          Compute the conditional alignment loss Eq. (17) ;

6          Compute the task-specific loss Eq. (20) ;

7          Optimize $(g, h)$ by objective Eq. (19) ;

8       **end**

9    **end**

10   **Output:** Adapted target model $(g^*, h^*)$.

---

Emprically, $N_2$ is set to 5 and $N_1$ depends on the size of target dataset. The primary computational load comes from the calculation of the exponential functions in Eq. (14) and Eq. (17). In commonly used scientific computing packages, this calculation process [28] can be employed to reduce the computational load.

### A.3 Influence of target training data size

As mentioned in Sec. 4, we used 4500 images in $\mathcal{D}_M$ and 1667 images in $\mathcal{D}_D$ for cross-domain learning in each task. Here we provide the number of target domain samples used by various DA methods in our comparisons:

| Method | DAGEN | PnP-GA | GSA-Gaze | HFC | DCUA | PnP-GA+ |
|---|---|---|---|---|---|---|
| Num. | 500 | <100 | 1000 | 100 | 100 | <100 |

Table 7: Number of target domain samples used by various DA methods

While the amount of unlabeled target-domain data we use may be larger than that used in some DG or UDA methods, we note the following:

- The unlabeled data is inexpensive and easy to collect in real-world scenarios (e.g., by simply capturing images with a camera), whereas labeled data requires careful calibration and annotation. Therefore, **using slightly more unlabeled samples does not significantly increase practical costs**. Besides, their effective contribution to training is much lower than that of labeled data, so small differences in sample size have limited influence on performance.

- Many existing approaches implicitly increase the diversity and amount of target-domain data via aggressive data augmentation, style transfer, or synthetic sample generation. Note that DG methods do not rely on target domain samples, but often heavily utilize **data augmentation or auxiliary tasks** to simulate domain shifts implicitly. While such methods implicitly enhance the effective size of training data, our method does not employ such augmentation strategies.

To address possible concern that performance gain might come from using more data, we conducted an experiment using only 100 unlabeled target samples, and report results below: Although performance degrades slightly with fewer samples, our method still achieves SOTA performance. It's also important to note that PnP-GA+, while using fewer samples, relies heavily on extensive data augmentations and

| Method | $\mathcal{D}_E \to \mathcal{D}_M$ | $\mathcal{D}_E \to \mathcal{D}_D$ | $\mathcal{D}_G \to \mathcal{D}_M$ | $\mathcal{D}_G \to \mathcal{D}_D$ | Avg |
|---|---|---|---|---|---|
| ResNet-18 | 8.05 | 9.03 | 7.41 | 8.83 | 8.33 |
| GLSGE w/ $n^t > 1000$ | **5.31** | 6.21 | **5.43** | **7.30** | **6.06** |
| GLSGE w/ $n^t = 100$ | 5.47 | 6.38 | 5.65 | 7.41 | 6.23 |
| PnP-GA+ | 5.34 | **5.73** | 6.10 | 7.62 | 6.20 |

Table 8: Comparision using only 100 unlabeled target samples with ResNet-18 as backbone model.

up to 10 auxiliary models. In contrast, our method uses no data augmentation or auxiliary modules, making our results more interpretable and directly attributable to the core methodology.

Finally, we briefly explain why our method remains effective even with fewer target samples. It's note that unlabeled data provides distributional information rather than supervision. In our method, label shift correction does not involve learning process and requires only a moderate number of samples for direct estimation. Conditional shift correction relies on feature alignment, which is more sample-sensitive and accounts for most of the performance drop. Nonetheless, even with a limited number of samples, our method achieves competitive results.

### A.4 Implementation details

For experiments using ResNet-18 and ResNet-50 as backbones models, the ResNet model pre-trained on ImageNet is assembled with a two-layers MLP as shallow feature extractor and a linear layer as the gaze predictor. The deep backbone module of ResNet is frozen in cross-domain training process. We use the Adam optimizer with the learning rate of $3e - 5$ and a cosine annealing scheduler to decrease the learning rate in the training process. The batch size is set to be 100. As the domain shift is distinct at the beginning, we alternately correct the label shift and the conditional shift to produce better pseudo label. The confidence that decides the truncated area in label shift correction process is emprically set to 0.7 for all tasks.

For experiments using GazeTR as backbone models, the hybrid GazeTR is pretrained in $\mathcal{D}_E$. All training parameters are consistent with the settings in original paper [6]. For experiments using FSCI as backbone models, we directly download the pretrained models open-sourced by the authors. Similarly, we froze their deep backbone module in cross-domain training process and assemble a shallow MLP model as in ResNet models.

An NVIDIA RTX 4080 GPU is used for the experiments. The mean results of five times random experiments are reported.

## B    Further discussions on factorization of joint distribution

In 3.1, we mentioned the factorization of joint distribution $P_{X,Y}(x, y)$ and its implications for CDGE. Here we provide a more detailed explanation. Specifically we adopted the factorization $P_{X,Y}(x, y) = P_{X|Y}(x|y)P_Y(y)$, because:

- $P(Y)$ reflects the distribution of gaze directions in the population, which is naturally subject to domain-specific priors (e.g., gaze behavior differs between indoor vs. outdoor settings). So the label shift naturally arises in CDGE.
- $P(X|Y)$ represents how visual observations (images) are generated given a particular gaze direction, capturing domain-specific appearance variations (e.g., lighting, head pose, background), which significantly affects the predictor behavior.

In contrast, the alternative factorization $P(Y|X)P(X)$ presents several limitations in our setting:

- Aligning the marginal distribution $P(X)$ is less informative, as it mixes samples from all labels and may not preserve task-relevant structure.
- Domain shift in $P(Y|X)$, i.e., the posterior distribution, lacks clear, observable semantic meaning and is harder to control or interpret in practice.

Therefore, we choose the $P(X|Y)P(Y)$ formulation as it better reflects the generative structure of the data in gaze estimation and provides clearer insight into the nature of domain shift in this task.

## C  PCOD

We consider the estimation of PCOD as two parts: first-order and second order statistics terms

$$d_{\text{COD}}^2(P_{X|Y}^s, P_{X|Y}^t) = \underbrace{\|\mathcal{U}_{X|Y}^s - \mathcal{U}_{X|Y}^t\|_{\mathcal{H}_{\mathcal{K}}}^2}_{\text{first-order term}} + \underbrace{\text{tr}(\mathcal{C}_{XX|Y}^{ss} + \mathcal{C}_{XX|Y}^{tt} - 2\mathcal{C}_{XX|Y}^{st})}_{\text{second-order term}}, \qquad (21)$$

Let $(\mathbf{x}_i^s, \mathbf{y}_i^s)_{i=1}^{n_s}$ and $(\mathbf{x}_i^t, \mathbf{y}_i^t)_{i=1}^{n_t}$ be two set of samples drawn i.i.d. from source and target domain. For simplicity, $n_s$ and $n_t$ are both set to $n$. In kernel method, samples are mapped to RKHS $\mathcal{H}_{\mathcal{X}} \oplus \mathcal{H}_{\mathcal{Y}}$ by the implicit feature map $(\phi, \psi)$. Denote the feature map matrices by $(\mathbf{\Phi}^s, \mathbf{\Psi}^s)$ and $(\mathbf{\Phi}^t, \mathbf{\Psi}^t)$, the explicit kernel matrices $\mathbf{K}_X^{ss} = \mathbf{\Phi}^s \mathbf{\Phi}^{sT}$, $\mathbf{K}_Y^{ss} = \mathbf{\Psi}^s \mathbf{\Psi}^{sT}$ are computed as $(\mathbf{K}_X^{ss})_{ij} = k_{\mathcal{X}}(\mathbf{x}_i^s, \mathbf{x}_j^s)$, $(\mathbf{K}_Y^{ss})_{ij} = k_{\mathcal{Y}}(\mathbf{y}_i^s, \mathbf{y}_j^s)$, respectively. And so as $\mathbf{K}_X^{tt}$, $\mathbf{K}_X^{st}$ and $\mathbf{K}_Y^{tt}$.

Let $\mathbf{1}_n$ be the n-dimensional vector with all elements equal to 1 and $\mathbf{I}_n = diag(\mathbf{1}_n)$ be the n-dimensional identity matrix, $\mathbf{q} = (q_1, \ldots, q_n)^T$ be the discretization of the target probability density function $p_Y^t(y)$ and $\mathbf{Q} = diag(\mathbf{q})$ be the diagonal matrix induced by $\mathbf{q}$. The empirical centering matrix can be defined as $\mathbf{H}_n = \mathbf{I}_n - \frac{1}{n}\mathbf{1}_n\mathbf{1}_n^T$. And the probability-aware centering matrix depending on $\mathbf{q}$ can be defined as $\mathbf{H}_q = \mathbf{Q} - \mathbf{q}\mathbf{q}^T$. Then the centralized kernel matrix is defined as $\mathbf{G}_Y = \mathbf{H}_n \mathbf{K}_Y \mathbf{H}_n$ and $\mathbf{G}_{Y^\omega} = \mathbf{B}^T \mathbf{K}_Y \mathbf{B}$, where $\mathbf{B} = (\sqrt{\mathbf{Q}} - \mathbf{q}\sqrt{\mathbf{q}}^T)$ satisfies $\mathbf{H}_q = \mathbf{B}\mathbf{B}^T$.

### C.1  First-order term of PCOD

With above notations, the probability-aware covariance operator $\hat{\mathcal{C}}_{XY^\omega}$ can be estimated as $\hat{\mathcal{C}}_{XY^\omega} = \mathbf{\Phi}\mathbf{Q}\mathbf{\Psi}^T$. Then the conditional mean operator can be estimated as

$$\begin{aligned}
\hat{\mathcal{U}}_{X|Y^\omega} &= \hat{\mathcal{C}}_{XY^\omega}(\hat{\mathcal{C}}_{Y^\omega Y^\omega} + \varepsilon\mathbf{I})^{-1} \\
&= \mathbf{\Phi}\mathbf{Q}\mathbf{\Psi}^\top(\mathbf{\Psi}\mathbf{Q}\mathbf{\Psi}^\top + \varepsilon\mathbf{I})^{-1} \\
&= \mathbf{\Phi}\mathbf{Q}(\mathbf{K}_Y\mathbf{Q} + \varepsilon\mathbf{I})^{-1}\mathbf{\Psi}^\top,
\end{aligned} \qquad (22)$$

the last equation is deduced by the following equations:

$$\begin{aligned}
\mathbf{\Psi}^\top(\mathbf{\Psi}\mathbf{Q}\mathbf{\Psi}^\top + \varepsilon\mathbf{I}) &= \mathbf{\Psi}^\top\mathbf{\Psi}\mathbf{Q}\mathbf{\Psi}^\top + \varepsilon\mathbf{\Psi}^\top \\
&= (\mathbf{\Psi}^\top\mathbf{\Psi}\mathbf{Q} + \varepsilon\mathbf{I})\mathbf{\Psi}^\top
\end{aligned} \qquad (23)$$

and

$$\mathbf{\Psi}^\top(\mathbf{\Psi}\mathbf{Q}\mathbf{\Psi}^\top + \varepsilon\mathbf{I})^{-1} = (\mathbf{\Psi}^\top\mathbf{\Psi}\mathbf{Q} + \varepsilon\mathbf{I})^{-1}\mathbf{\Psi}^\top. \qquad (24)$$

The the HS-norm of the condition mean operator can be calculated as

$$\begin{aligned}
\|\hat{\mathcal{U}}_{X|Y^\omega}^s\|_{HS} &= \text{tr}[\mathbf{\Phi}\mathbf{Q}(\mathbf{K}_Y\mathbf{Q} + \varepsilon\mathbf{I})^{-1}\mathbf{\Psi}^\top\mathbf{\Psi}(\mathbf{K}_Y\mathbf{Q} + \varepsilon\mathbf{I})^{-1^T}\mathbf{Q}^\top\mathbf{\Phi}^\top] \\
&= \text{tr}[\mathbf{K}_X^{ss}\mathbf{Q}(\mathbf{K}_Y^{ss}\mathbf{Q} + \varepsilon\mathbf{I})^{-1}\mathbf{K}_Y^{ss}(\mathbf{Q}\mathbf{K}_Y + \varepsilon\mathbf{I})^{-1}\mathbf{Q}].
\end{aligned} \qquad (25)$$

By replacing $\mathbf{Q}$ as the empirical $\frac{1}{n}\mathbf{I}$, the estimation for target domain is derived as

$$\begin{aligned}
\|\mathcal{U}_{X|Y}^t\|_{HS} &= \text{tr}\left(\mathbf{\Phi}^t(\mathbf{K}_Y^{tt} + \varepsilon n\mathbf{I})^{-1}\mathbf{\Psi}^{t\top}\mathbf{\Psi}^t(\mathbf{K}_Y^{tt} + \varepsilon n\mathbf{I})^{-1}\mathbf{\Phi}^{t\top}\right) \\
&= \text{tr}\left(\mathbf{K}_X^{tt}(\mathbf{K}_Y^{tt} + \varepsilon n\mathbf{I})^{-1}\mathbf{K}_Y^{tt}(\mathbf{K}_Y^{tt} + \varepsilon n\mathbf{I})^{-1}\right).
\end{aligned} \qquad (26)$$

The inner product term is then calculated as

$$\begin{aligned}
\langle\mathcal{U}_{X|Y^\omega}^s, \mathcal{U}_{X|Y}^t\rangle &= \text{tr}[\mathbf{\Phi}^s\mathbf{Q}(\mathbf{K}_Y^{ss}\mathbf{Q} + \varepsilon\mathbf{I})^{-1}\mathbf{\Psi}^{s\top}\mathbf{\Psi}^t(\mathbf{K}_Y^{tt} + \varepsilon n\mathbf{I})^{-1}\mathbf{\Psi}^{t\top}] \\
&= \text{tr}\left(\mathbf{K}_X^{ts}\mathbf{Q}(\mathbf{K}_Y^{ss}\mathbf{Q} + \varepsilon\mathbf{I})^{-1}\mathbf{K}_Y^{st}(\mathbf{K}_Y^{tt} + \varepsilon n\mathbf{I})^{-1}\right).
\end{aligned} \qquad (27)$$

Then the first part of PCOD is derived:

$$\begin{aligned}
d_{\text{CMMD}^\omega}^2 &= \text{tr}\left(\mathbf{K}_X^{ss}\mathbf{Q}(\mathbf{K}_Y^{ss}\mathbf{Q} + \varepsilon\mathbf{I})^{-1}\mathbf{K}_Y^{ss}(\mathbf{Q}\mathbf{K}_Y^{ss} + \varepsilon\mathbf{I})^{-1}\mathbf{Q}\right) \\
&\quad + \text{tr}\left(\mathbf{K}_X^{tt}(\mathbf{K}_Y^{tt} + \varepsilon n\mathbf{I})^{-1}\mathbf{K}_Y^{tt}(\mathbf{K}_Y^{tt} + \varepsilon n\mathbf{I})^{-1}\right) \\
&\quad - 2\,\text{tr}\left(\mathbf{K}_X^{ts}\mathbf{Q}(\mathbf{K}_Y^{ss}\mathbf{Q} + \varepsilon\mathbf{I})^{-1}\mathbf{K}_Y^{st}(\mathbf{K}_Y^{tt} + \varepsilon n\mathbf{I})^{-1}\right).
\end{aligned} \qquad (28)$$

## C.2 Second-order term of PCOD

For the second-order terms, the cross conditional operator can be estimated as

$$
\begin{aligned}
\hat{\mathcal{C}}_{XX|Y^\omega} &= \hat{\mathcal{C}}_{XX} - \hat{\mathcal{C}}_{XY^\omega}(\hat{\mathcal{C}}_{YY^\omega} + \varepsilon \mathbf{I})^{-1}\hat{\mathcal{C}}_{Y^\omega X} \\
&= \mathbf{\Phi}\mathbf{H}_q\mathbf{\Phi}^T - \mathbf{\Phi}\mathbf{H}_q\mathbf{\Psi}^T(\mathbf{\Psi}\mathbf{H}_q\mathbf{\Psi}^T + \varepsilon \mathbf{I})^{-1}\mathbf{\Psi}\mathbf{H}_q\mathbf{\Phi}^T \\
&= \mathbf{\Phi}\mathbf{B}\left[\mathbf{I}_n - \mathbf{B}^T\mathbf{\Psi}^T(\mathbf{\Psi}\mathbf{B}(\mathbf{\Psi}\mathbf{B})^T + \varepsilon \mathbf{I})^{-1}\mathbf{\Psi}\mathbf{B}\right]\mathbf{B}^T\mathbf{\Phi}^T
\end{aligned}
\tag{29}
$$

By SWM formula [36] the inverse term can be calculated as

$$
\begin{aligned}
(\hat{\mathcal{C}}_{YY^\omega} + \varepsilon \mathbf{I})^{-1} &= (\varepsilon \mathbf{I} + \mathbf{\Psi}\mathbf{B}(\mathbf{\Psi}\mathbf{B})^T)^{-1} \\
&= \frac{1}{\varepsilon}\left[\mathbf{I}_D - \mathbf{\Psi}\mathbf{B}(\varepsilon \mathbf{I}_n + \mathbf{B}^T\mathbf{K}_Y\mathbf{B})^{-1}\mathbf{B}^T\mathbf{\Psi}^T\right] \\
&= \mathbf{\Phi}\mathbf{B}\left[\mathbf{I}_n - \frac{1}{\varepsilon}(\mathbf{B}^T\mathbf{K}_Y\mathbf{B} - \mathbf{B}^T\mathbf{K}_Y\mathbf{B}(\varepsilon \mathbf{I}_n + \mathbf{B}^T\mathbf{K}_Y\mathbf{B})^{-1}\mathbf{B}^T\mathbf{K}_Y\mathbf{B})\right]\mathbf{B}^T\mathbf{\Phi}^T \\
&= \mathbf{\Phi}\mathbf{B}\left[\mathbf{I}_n - \frac{1}{\varepsilon}(\mathbf{G}_{Y^\omega} - \mathbf{G}_{Y^\omega}(\varepsilon \mathbf{I}_n + \mathbf{G}_{Y^\omega})^{-1}\mathbf{G}_{Y^\omega})\right]\mathbf{B}^T\mathbf{\Phi}^T
\end{aligned}
\tag{30}
$$

For the term $\mathbf{I}_n - \frac{1}{\varepsilon}(\mathbf{G}_{Y^\omega} - \mathbf{G}_{Y^\omega}(\varepsilon \mathbf{I}_n + \mathbf{G}_{Y^\omega})^{-1}\mathbf{G}_{Y^\omega})$, we perform eigenvalue decomposition on $\mathbf{G}_{Y^\omega}$ such that $G_{Y^\omega} = \mathbf{U}\mathbf{D}\mathbf{U}^T$, where $\mathbf{U}$ is orthogonal and $\mathbf{D}$ is diagonal with $d_i \geq 0$. Then we can simplify the term as

$$
\begin{aligned}
\mathbf{I}_n &- \frac{1}{\varepsilon}(\mathbf{G}_{Y^\omega} - \mathbf{G}_{Y^\omega}(\varepsilon \mathbf{I}_n + \mathbf{G}_{Y^\omega})^{-1}\mathbf{G}_{Y^\omega}) \\
&= \mathbf{I}_n - \frac{1}{\varepsilon}\left[\mathbf{U}\mathbf{D}\mathbf{U}^T - \mathbf{U}\mathbf{D}\mathbf{U}^T(\varepsilon \mathbf{I}_n + \mathbf{U}\mathbf{D}\mathbf{U}^T)^{-1}\mathbf{U}\mathbf{D}\mathbf{U}^T\right] \\
&= \mathbf{I}_n - \frac{1}{\varepsilon}\left[\mathbf{U}\mathbf{D}\mathbf{U}^T - \mathbf{U}\mathbf{D}(\varepsilon \mathbf{I}_n + \mathbf{D})^{-1}\mathbf{D}\mathbf{U}^T\right] \\
&= \mathbf{U}\left[\mathbf{I}_n - \frac{1}{\varepsilon}(\mathbf{D} - \mathbf{D}(\varepsilon \mathbf{I}_n + \mathbf{D})^{-1}\mathbf{D})\right]\mathbf{U}^T \\
&=: \mathbf{U}\mathbf{D}'\mathbf{U}^T,
\end{aligned}
\tag{31}
$$

where $\mathbf{D}'$ is diagonal and

$$
d_i' = 1 - \frac{1}{\varepsilon}(d_i - \frac{d_i^2}{d_i + \varepsilon}) = \frac{\varepsilon}{d_i + \varepsilon} > 0.
\tag{32}
$$

Finally, it is derived as

$$
\mathbf{I}_n - \frac{1}{\varepsilon}(\mathbf{G}_{Y^\omega} - \mathbf{G}_{Y^\omega}(\varepsilon \mathbf{I}_n + \mathbf{G}_{Y^\omega})^{-1}\mathbf{G}_{Y^\omega}) = \varepsilon(\mathbf{G}_{Y^\omega} + \varepsilon \mathbf{I}_n)^{-1}.
\tag{33}
$$

Then the conditional covariance operator is simplified as

$$
\begin{aligned}
\hat{\mathcal{C}}_{XX|Y^\omega} &= \mathbf{\Phi}\mathbf{B}[\mathbf{I}_n - \frac{1}{\varepsilon}(\mathbf{G}_{Y^\omega} - \mathbf{G}_{Y^\omega}(\varepsilon \mathbf{I}_n + \mathbf{G}_{Y^\omega})^{-1}\mathbf{G}_{Y^\omega})]\mathbf{B}^T\mathbf{\Phi}^T \\
&=: \mathbf{\Phi}\mathbf{B}\mathbf{L}^\omega\mathbf{B}^T\mathbf{\Phi}^T \\
&= \mathbf{\Phi}\mathbf{B}\varepsilon(\mathbf{G}_{Y^\omega} + \varepsilon \mathbf{I}_n)^{-1}\mathbf{B}^T\mathbf{\Phi}^T \\
&= \mathbf{\Phi}\mathbf{B}\varepsilon(\mathbf{G}_{Y^\omega} + \varepsilon \mathbf{I}_n)^{-1}\mathbf{B}^T\mathbf{\Phi}^T,
\end{aligned}
\tag{34}
$$

where $\mathbf{L}^\omega := \varepsilon(\mathbf{G}_{Y^\omega} + \varepsilon \mathbf{I}_n)^{-1}$. And the cross conditional covariance operator is then calculated as

$$
\begin{aligned}
\mathcal{C}_{XX|Y^\omega}^{st} &= \sqrt{\sqrt{\mathcal{C}_{XX|Y^\omega}^s}\mathcal{C}_{XX|Y}^t\sqrt{\mathcal{C}_{XX|Y^\omega}^s}} \\
&= \sqrt{\sqrt{\mathbf{\Phi}^s\mathbf{B}\mathbf{L}^\omega\mathbf{B}^T\mathbf{\Phi}^{s^T}}\frac{1}{n}(\mathbf{\Phi}^t\mathbf{H}_n\mathbf{L}\mathbf{H}_n\mathbf{\Phi}^t)\sqrt{\mathbf{\Phi}^s\mathbf{B}\mathbf{L}^\omega\mathbf{B}^T\mathbf{\Phi}^{s^T}}} \\
&= \frac{1}{\sqrt{n}}\sqrt{\sqrt{\mathbf{\Phi}^s\mathbf{M}^\omega\mathbf{M}^{\omega T}\mathbf{\Phi}^{s^T}}\mathbf{\Phi}^t\mathbf{M}(\mathbf{\Phi}^t\mathbf{M})^T\sqrt{\mathbf{\Phi}^s\mathbf{M}^\omega\mathbf{M}^{\omega T}\mathbf{\Phi}^{s^T}}} \\
&= \frac{1}{\sqrt{n}}\sqrt{(\sqrt{\mathbf{\Phi}^s\mathbf{M}^\omega\mathbf{M}^{\omega T}\mathbf{\Phi}^{s^T}}\mathbf{\Phi}^t\mathbf{M})(\sqrt{\mathbf{\Phi}^s\mathbf{M}^\omega\mathbf{M}^{\omega T}\mathbf{\Phi}^{s^T}}\mathbf{\Phi}^t\mathbf{M})^T}
\end{aligned}
\tag{35}
$$

So the trace term is

$$
\begin{aligned}
\mathrm{tr}(\mathcal{C}^{st}_{XX|Y^\omega}) &= \frac{1}{\sqrt{n}}\mathrm{tr}\sqrt{(\sqrt{\boldsymbol{\Phi}^s\mathbf{M}^\omega\mathbf{M}^{\omega T}\boldsymbol{\Phi}^{s T}}\boldsymbol{\Phi}^t\mathbf{M})^T(\sqrt{\boldsymbol{\Phi}^s\mathbf{M}^\omega\mathbf{M}^{\omega T}\boldsymbol{\Phi}^{s T}}\boldsymbol{\Phi}^t\mathbf{M})} \\
&= \frac{1}{\sqrt{n}}\mathrm{tr}\sqrt{\mathbf{M}^T\boldsymbol{\Phi}^{t T}(\boldsymbol{\Phi}^s\mathbf{M}^\omega\mathbf{M}^{\omega T}\boldsymbol{\Phi}^{s T})\boldsymbol{\Phi}^t\mathbf{M}} \\
&= \frac{1}{\sqrt{n}}\mathrm{tr}\sqrt{(\mathbf{M}^T\mathbf{K}^{ts}_X\mathbf{M}^\omega)\,(\mathbf{M}^T\mathbf{K}^{ts}_X\mathbf{M}^\omega)^T} \\
&= \frac{1}{\sqrt{n}}\|\mathbf{M}^T\mathbf{K}^{ts}_X\mathbf{M}^\omega\|_*
\end{aligned}
\tag{36}
$$

where the $\mathbf{M}^\omega$ and $\mathbf{M}^\omega$ are defined by $\mathbf{H}_n\mathbf{L}\mathbf{H}_n = \mathbf{M}\mathbf{M}^T$ and $\mathbf{B}\mathbf{L}^\omega\mathbf{B}^T = \mathbf{M}^\omega\mathbf{M}^{\omega T}$.

Summarizing the results on conditional operator, the second part of PCOD is then derived:

$$
\hat{d}^2_{CKB^\omega} = \varepsilon\mathrm{tr}\left(\mathbf{G}^s_{X^\omega}(\mathbf{G}^s_{Y^\omega} + \varepsilon\mathbf{I}_n)^{-1}\right) + \varepsilon\mathrm{tr}\left(\mathbf{G}^t_X(\mathbf{G}^t_Y + \varepsilon n\mathbf{I}_n)^{-1}\right) - \frac{2}{\sqrt{n}}\|\mathbf{M}^T\mathbf{K}^{ts}_X\mathbf{M}^\omega\|_*.
\tag{37}
$$

