# OpenReview forum: "A Generalized Label Shift Perspective for Cross-Domain Gaze Estimation"
_NeurIPS.cc/2025/Conference — NeurIPS 2025 poster_

### Official Review · Reviewer_nmUr · 2025-06-07

**Clarity:** 3
**Significance:** 3
**Originality:** 3
**Rating:** 4
**Confidence:** 2

**Summary:**

This paper proposes a method for cross-domain gaze estimation (CDGE), addressing key limitations of existing domain adaptation methods for label shift and conditional shift. It introduces the Generalized Label Shift (GLS) theory as the solution. Experiments between several datasets demonstrate the superior generalizability compared to existing methods.

**Questions:**

See the weaknesses

**Ethical Concerns:**

["NO or VERY MINOR ethics concerns only"]

**Final Justification:**

According to the rebuttal, my main concerns were addressed. The remaining problem is that the proposed method in this work seems to be a general solution for the label and conditional shift problems for continuous labels, which is loosely coupled with the task of gaze estimation. It's better to include a discussion about its relationship to other tasks with continuous labels and the potential to apply the method to other fields to extend the impact of this work. Given the strong performance and the theoretical support, I'll keep my positive rating of borderline accept.

**Limitations:**

yes

**Quality:**

3

**Strengths And Weaknesses:**

Strengths:
- Extensive experiments demonstrate its superior performance on various datasets.
- Sufficient theoretical support provided.

Weaknesses:
- While the abstract and introduction clearly state the contributions, the motivation for addressing Generalized Label Shift (GLS) in gaze estimation could be better justified. The authors should explicitly explain why existing domain-invariant methods fail in gaze estimation compared to other tasks. Current explanations mainly focus on the general difference between the proposed method and previous works, but are not related to task-specific problems.

- It should also be explained why label shift and conditional shift are significant and inherent problems for gaze estimation and differ from in other tasks. In the descriptions of L128-137, the involved problems can also appear in other tasks like object detection and categorization. How Sec. 3.4 solves the problems specifically in gaze estimation is also unclear, which is more like a general solution. It's better to make the relationship between the task-specific problem analysis and the proposed solution more clearly.

- Typos. e.g. L18, "appereance-based"

---

> ### Author Rebuttal · Authors · 2025-07-29
>
> Thank you for taking the time to review our paper and provide professional and constructive reviews. We are encouraged by the positive comments on the extensive experiments and the theoretical support. All suggestions will be carefully incorporated into final version.
>
> **_Q: Why existing domain-invariant methods fail in gaze estimation compared to other tasks_**
>
> Existing domain-invariant methods primarily address the covariate or conditional shift, assuming that the label distribution remains the same across domains. However, in CDGE problem, we observe that both label and conditional shift naturally occur, as discussed in Sec. 3.1, and illustrated in Fig. 2. Moreover, in **lines 162–165** of Sec. 3.2, we explicitly state:
>
> **"Previous works are essentially connected with invariant representation learning and omit the label shift correction. According to the GLS theory, such methods are insufficient for successful cross-domain learning when label shift exists."**
>
> Therefore, domain-invariant methods are insufficient for the CDGE problem. Our work addresses this limitation by introducing a GLS correction framework that jointly models and corrects the label and conditional shift.
>
> **_Q: Why label shift and conditional shift are significant and inherent problems for gaze estimation and differ from in other tasks._**
>
> (1) Why label and conditional shift are significant and inherent in gaze estimation:
>
> As described in Eq. (4), the core objective in CDGE is to minimize the target error by reducing the source error while correcting the domain shift. In **lines 96–108**, we introduce the GLS theory, which rigorously establishes the sufficiency and necessity of correcting both label and conditional shift for cross-domain learning. We further demonstrate in **lines 128–137** and **Fig. 2** that both forms of shift are naturally present in CDGE.
>
> Therefore, the label and conditional shift are inherent and significant in CDGE. We will highlight this conclusion more explicitly at the end of Sec. 3.1 in the final version.
>
> (2) Why this differs from other tasks:
>
> As discussed in **lines 96–111**, the GLS framework is general and applicable to various domain adaptation problems, including object detection and classification. However, existing GLS methods are primarily designed for classification tasks with finite discrete labels.
>
> In contrast, gaze estimation involves continuous-valued labels, making label and conditional shift correction significantly more challenging. This distinction is elaborated in **lines 170–174 and 190–193**, where we explain the difficulty of modeling continuous label distributions and designing appropriate conditional alignment strategies.
>
> Due to these challenges, existing GLS methods are not directly applicable to CDGE. In response, our work (Sec. 3.3 to 3.5) proposes a feasible GLS correction implementation which explicitly integrates the task-specific prior of gaze estimation.
>
> **_Q: How Sec. 3.4 solves the problems specifically in gaze estimation._**
>
> Sec. 3.4 introduces the PCOD method, which is motivated by the distributional properties of gaze estimation.
>
> As discussed in **lines 190–193**, most existing conditional alignment methods are designed for classification tasks with discrete labels, making them not applicable for gaze estimation with continuous labels. Then we consider the COD method as a starting point, which is more suitable for regression settings.
>
> Furthermore, in **lines 198–205**, we identify a critical limitation of COD under label shift: COD fails to embed conditional distribution with respect to a reweighted label distribution. To address this, we derive the Probability-aware COD (PCOD), which incorporates the learned reweighting function ω(y) into the alignment process, making it suitable for CDGE problem.
>
> Therefore, PCOD is applicable to other tasks with similar distributional properties, providing a more general-purpose solution within the GLS correction framework. Of course, beyond distributional aspects, gaze estimation has other unique properties, such as rotational equivariance. Integrating these domain-specific priors into the GLS correction method is an interesting direction for future work.
>
> **_Q: Typo._**
>
> We sincerely appreciate the reviewer's careful reading. All reported typos will be corrected in the final manuscript, and we will conduct a full proofread to ensure textual accuracy throughout the paper.

---

> > ### Comment · Reviewer_nmUr · 2025-08-03
> >
> > Thanks for the responses. According to the rebuttal, my main concerns were addressed. The remaining problem is that the proposed method in this work seems to be a general solution for the label and conditional shift problems for continuous labels, which is loosely coupled with the task of gaze estimation. It's better to include a discussion about its relationship to other tasks with continuous labels and the potential to apply the method to other fields to extend the impact of this work. I'll keep my positive rating of borderline accept.

---

> > > ### Author Response · Authors · 2025-08-04
> > >
> > > Thank you for your timely response and for acknowledging that our rebuttal addressed your main concerns. We now aim to clarify the remaining point regarding the **generality and task-specific relevance** of our proposed method.
> > >
> > > We agree with your observation that the proposed framework is general and can be applied to other tasks with continuous labels under label and conditional shift. In fact, this generality is by design: our method models CDGE as a GLS problem and introduces a GLS correction framework accordingly. This modeling paradigm is not limited to gaze estimation and can be potentially extended to a broad range of tasks with similar distributional properties, e.g., pose estimation.
> > >
> > > Meanwhile, we would also like to emphasize that **our contribution is not only in the formulation of a general framework**, but also in the design of **a task-specific implementation** that makes the GLS framework feasible and effective for CDGE. We believe this implementation step is **crucial** when adapting the general GLS framework to different tasks, and it often requires **task-specific insights** and **custom solutions**.
> > >
> > > We will include a discussion in the final version to clarify both the applicability of our framework to other continuous-label tasks and the importance of incorporating task-specific properties when applying it beyond gaze estimation.
> > >
> > > **Thank you again for your helpful comments and positive attitude.**

---

### Official Review · Reviewer_ZkHw · 2025-06-23

**Clarity:** 3
**Significance:** 3
**Originality:** 3
**Rating:** 4
**Confidence:** 4

**Summary:**

This paper addresses the cross-domain gaze estimation (CDGE) problem by introducing a novel perspective based on Generalized Label Shift (GLS) theory, modeling the domain shift as both label and conditional shifts. It proposes an importance reweighting strategy using a truncated Gaussian distribution to effectively handle the continuity challenges in label shift correction. Extensive experiments demonstrate that the method achieves superior generalization across domains and is applicable to various backbone models.

**Questions:**

Check Cons above

**Ethical Concerns:**

["NO or VERY MINOR ethics concerns only"]

**Final Justification:**

After checking the reviews and rebuttals, I think the authors have addressed most concerns well. Even though I still have a small concern about the fairness of the comparison, I accept the explanation in the rebuttal. I will keep my original rating of  Borderline accept.

**Limitations:**

Yes

**Quality:**

3

**Strengths And Weaknesses:**

Pros:
1. The paper proposes a new perspective for the domain adaptation in gaze estimation which is interesting.
2. This paper is theoretically sound.
3. Experiments are extensive and support the conclusion well.

Cons:
1. The proposed method seems effective but complicated. I am curious about the time cost for the adaptation process.
2. Is it too simple to modelize the gaze label distribution as a bivariate truncated Gaussian distribution? Do you have better choices?
3. PXY = PX|Y*PY or PY|X*PX. Does your approach also work for PY|X*PX? Why not choose PY|X*PX?
4. Figure 1 is not clear for illustrating the difference among different methods. I advise to including three subfigures to show the main ideas of the three methods.
5. How many images from the target domain are used for domain adaptation? It should be included in Table 2 and 3.

---

> ### Author Rebuttal · Authors · 2025-07-30
>
> Thank you for taking the time to review our paper and provide professional and constructive reviews. We are encouraged by the positive comments on the clear paper writing, limitations outlining and extensive experiment. All suggestions will be carefully incorporated into final version.
>
> **_Q: Better choice to modelize the gaze label distribution._**
>
> At its core, we aim to model a continuous function $f(y)$ that explicitly represents either the reweighting function ω(y) or the target-domain label distribution $p^t(y)$. The function $f$ should satisfy the following properties:
>
> -- Since gaze is a continuous variable, $f(y)$ should be defined over a continuous space so that for any $x∈D$, we can compute $y=f(x)$.
>
> -- Because the gaze range is bounded, the support of $f$ should also be compact. This can be naturally enforced by using truncated distributions.
>
> **Under these constraints, there are many possible modeling choices beyond a Gaussian—such as exponential family distributions, uniform distributions, or even shallow neural networks.** The choice of $f$ can be adapted to the characteristics of the specific task or dataset, which we believe is a promising direction for future work.
>
> In our current paper, as stated in **lines 175–180**, we adopt a bivariate truncated Gaussian for $f$ due to its simplicity, analytic tractability, and because it aligns reasonably well with the empirical observation in the target domain.
>
> However, we acknowledge that in other settings—e.g., driver gaze datasets where gaze points tend to concentrate around a few specific directions (e.g., forward, mirrors)—a **truncated Gaussian mixture model** may provide a more accurate fit. Incorporating such prior knowledge into the choice of $f$ is indeed a meaningful extension, and we plan to explore this in future work.
>
> Due to the time constraints of the rebuttal phase, we include here a reference result using a linear combination of three uniform distributions for $f$ as a comparison:
>
> |      	  | $\mathcal{D}_E \to \mathcal{D}_D$ | $\mathcal{D}_G \to \mathcal{D}_D$ |
> |------------|------------|------|
> |GLSGE w/ Gau. | 6.21 | 7.30 |
> | GLSGE w/ Uni. | 6.02 | 6.58 |
> |DCUA  | 5.95 | 6.4 |
>
> **_Q: Decomposition of the joint distribution._**
>
> We thank the reviewer for this insightful question. From a probabilistic perspective, both factorizations of the joint distribution—i.e., P(X,Y)=P(X|Y)P(Y) or P(Y|X)P(X) —are mathematically valid. **However, we adopt the factorization P(X,Y)=P(X|Y)P(Y) because it provides a more intuitive and interpretable modeling perspective for the CDGE task.** Specifically:
>
> -- P(Y) reflects the distribution of gaze directions in the population, which is naturally subject to domain-specific priors (e.g., gaze behavior differs between indoor vs. outdoor settings). So the label shift naturally arises in CDGE.
>
> -- P(X|Y) represents how visual observations (images) are generated given a particular gaze direction, capturing domain-specific appearance variations (e.g., lighting, head pose, background), which significantly affects the predictor behavior.
>
> In contrast, the alternative factorization P(Y|X)P(X) presents several limitations in our setting:
>
> -- Aligning the marginal distribution P(X) is less informative, as it mixes samples from all labels and may not preserve task-relevant structure.
>
> -- Domain shift in P(Y|X), i.e., the posterior distribution, lacks clear, observable semantic meaning and is harder to control or interpret in practice.
>
> Therefore, we choose the P(X|Y)P(Y) formulation as it better reflects the generative structure of the data in gaze estimation and provides clearer insight into the nature of domain shift in this task.
>
> **_Q: The difference among different methods in Fig. 1._**
>
> Thank you for the helpful suggestion. We agree that presenting three separate subfigures could make the differences among the methods more visually explicit. However, due to **space constraints** and our intent to **emphasize the conceptual connections between the approaches**, we chose to present all three within a single integrated diagram in Fig. 1.
>
> To help distinguish the methods clearly, we adopted a combination of colors and line styles, as described in the figure caption:
>
> -- The black dashed line represents DG methods, which perform invariant representation learning using only labeled source-domain data.
>
> -- The black solid line denotes UDA methods, which perform invariant representation learning using both labeled source and unlabeled target data.
>
> -- The orange solid line represents our proposed GLS correction method, which not only performs invariant representation learning, but also explicitly corrects for label shift, and incorporates probability-aware modeling into both the representation and predictor learning.
>
> We acknowledge that the current figure may not fully convey these distinctions at a glance. In the final version, we will enhance the caption with a more detailed explanation and consider making visual refinements (e.g., clearer legends, annotations) to improve clarity. If space permits, we will also explore the possibility of using subfigures, as the reviewer suggests.
>
> **_Q: Number of target domain images used for domain adaptation._**
>
> During cross-domain learning, we use **10%** of the unlabeled target domain images for training and another **10%** for validation, with the remaining **80%** used for testing. Concretely, this corresponds to 4,500 unlabeled images in the DM dataset, and 1,667 unlabeled images in the DD dataset.
>
> While the amount of unlabeled target-domain data we use may be larger than that used in some DG or UDA methods, we note the following:
>
> -- **the unlabeled data is inexpensive and easy to collect in real-world scenarios** (e.g., by simply capturing images with a camera).
>
> -- **Many existing approaches implicitly increase the diversity and amount of target-domain data via aggressive data augmentation, style transfer, or synthetic sample generation.** While such methods implicitly enhance the effective size of training data, our method does not employ such augmentation strategies.
>
> We believe our experimental setting remains fair and reflects a practical and realistic scenario for cross-domain learning, and it highlights the benefit of our approach in leveraging readily available unlabeled data. We will clarify this setting more explicitly in the final version of the paper.

---

> > ### Comment · Reviewer_ZkHw · 2025-08-05
> >
> > I appreciate authors' responses. Regarding the number of target domain images used for domain adaptation, could you list the number of images used for the compared methods and your method in the experiment of the paper? You said the numbers are different for the methods, could you give the results using the same number of images for adaptation? I think using more images to get better results won't make your method more advantageous.

---

> > > ### Author Response · Authors · 2025-08-06
> > >
> > > Thank you for your thoughtful question. We fully understand your concern regarding the number of target domain samples used in training.
> > >
> > > First, we would like to clarify that our method only utilizes unlabeled target domain samples. **Compared to labeled samples**, unlabeled samples have the following characteristics:
> > >
> > > 1.	Their effective contribution to training is much lower than that of labeled data, so **small differences in sample size have limited influence on performance.**
> > >
> > > 2.	They are much easier to collect, especially in gaze estimation tasks—unlabeled data can be obtained by simply taking pictures, whereas labeled data requires careful calibration and annotation. Therefore, **using slightly more unlabeled samples does not significantly increase practical costs.**
> > >
> > > Next, we provide the number of target domain samples used by various DA methods in our comparisons:
> > >
> > > | DAGEN | PnP-GA | GSA-Gaze |  HFC   | DCUA  | PnP-GA+ |
> > > |--------|--------|----------|--------|--------|---------|
> > > | 500    |  <100  |  1000   |  100   |  100   |  <100   |
> > >
> > > Note that DG methods do not rely on target domain samples, but often heavily utilize **data augmentation or auxiliary tasks** to simulate domain shifts implicitly.
> > >
> > > To address your concern that performance gain might come from using more data, we conducted an experiment using only **100 unlabeled target samples**, and report results below (ResNet-18 backbone):
> > >
> > > |  | $D_E \to D_M$ | $D_E \to D_D$ | $D_G \to D_M$ | $D_G \to D_D$ | Avg. |
> > > |-------|--------------|-------------|--------------|--------------|-----|
> > > | GLSGE w/ $n^t >1000$ |    5.31      |     6.21     |     5.43     |    7.30      |  6.06 |
> > > | GLSGE w/ $n^t =100$ |   5.47       |    6.38      |    5.65      |    7.41      | 6.23 |
> > > | PnP-GA+ | 5.34      |     5.73      |    6.10       |    7.62      | 6.20 |
> > >
> > > Although performance degrades slightly with fewer samples, our method still achieves SOTA performance. It's also important to note that PnP-GA+, while using fewer samples, relies heavily on **extensive data augmentations** and up to **10 auxiliary models**. In contrast, our method uses **no data augmentation or auxiliary modules**, making our results more interpretable and directly attributable to the core methodology.
> > >
> > > Finally, we briefly explain why our method remains effective even with fewer target samples. It's note that **unlabeled data provides distributional information rather than supervision**. In our method, label shift correction does not involve learning process and requires only a moderate number of samples for direct estimation. Conditional shift correction relies on feature alignment, which is more sample-sensitive and accounts for most of the performance drop. Nonetheless, even with a limited number of samples, our method achieves competitive results.
> > >
> > > Thank you again for your valuable question. Please let us know if you have further concerns or suggestions.

---

> > > > ### Comment · Reviewer_ZkHw · 2025-08-07
> > > >
> > > > Thanks for the answer. My concern has been addressed.

---

> > > > > ### Author Response · Authors · 2025-08-07
> > > > >
> > > > > Thank you for your positive feedback. Your comments are greatly appreciated and will be carefully incorporated in the final revision of our paper.

---

> ### Comment · Area_Chair_bybh · 2025-08-05
> **Please discuss rebuttal with authors**
>
> Please discuss rebuttal with authors

---

### Official Review · Reviewer_bWiS · 2025-06-27

**Clarity:** 3
**Significance:** 3
**Originality:** 3
**Rating:** 4
**Confidence:** 4

**Summary:**

This paper focuses on Cross-domain Gaze Estimation (CDGE) task, which helps apply gaze estimation to various environments. This work introduce Generalized Label Shift (GLS) theory to gaze estimation, consisting of shifts in both the label and conditional distributions. Evaluations are conducted on different datasets and network structures, showing that the proposed method has achieved state-of-the-art performance.

**Questions:**

Please refer to the weaknesses.

**Ethical Concerns:**

["NO or VERY MINOR ethics concerns only"]

**Final Justification:**

Thanks for the responses. After reviewing the rebuttal and considering the comments from the other reviewers, I have decided to maintain my original rating as borderline accept.

**Quality:**

3

**Strengths And Weaknesses:**

Strength:
1. The tasking definition is clear and easy to understand its value.
2. The authors clearly outline limitations in existing GLS correction methods for gaze estimation and propose appropriate solutions.
3. Comprehensive experiments have been conducted, which helps confirm the method's effectiveness.

Weakness:
1. The rationale for modeling the label distribution as a bivariate truncated Gaussian is unclear and needs stronger justification.
2. The relationship between sections in Sec. 3 lacks clarity. Specifically, Sec. 3.2 appears to describe previous methods, making it more suitable for inclusion in the related work section. From my point of view, it would be useful to have an overview at the beginning of Sec. 3.
3. It should address how GLS correction methods have benefited other tasks and clarify their value specifically in gaze estimation. Additionally, an overview of the current state of gaze estimation research should be provided to better contextualize the contribution.

---

> ### Author Rebuttal · Authors · 2025-07-30
>
> Thank you for taking the time to review our paper and provide professional and constructive reviews. We are encouraged by the positive comments on the clear paper writing, limitations outlining and extensive experiment. All suggestions will be carefully incorporated into final version.
>
> **_Q: Why modeling the label distribution as a bivariate truncated Gaussian._**
>
> Thank you for the insightful question.
> At its core, we aim to model a continuous function $f(y)$ that explicitly represents either the reweighting function ω(y) or the target-domain label distribution $p^t(y)$. The function $f$ should satisfy the following properties:
>
> -- Since gaze is a continuous variable, $f(y)$ should be defined over a continuous space so that **for any $x∈D$, we can compute $y=f(x)$.**
>
> -- Because the gaze range is bounded, the support of $f$ should also be compact. This can be naturally enforced by using truncated distributions.
>
> Under these constraints, there are many possible modeling choices beyond a Gaussian—such as exponential family distributions, uniform distributions, or even shallow neural networks. The choice of $f$ can be adapted to the characteristics of the specific task or dataset, which we believe is a promising direction for future work.
>
> In our current paper, as stated in **lines 175–180**, we adopt a bivariate truncated Gaussian for $f$ due to its simplicity, analytic tractability, and because it aligns reasonably well with the empirical observation in the target domain.
>
> However, we acknowledge that in other settings—e.g., driver gaze datasets where gaze points tend to concentrate around a few specific directions (e.g., forward, mirrors)—a **truncated Gaussian mixture model** may provide a more accurate fit. Incorporating such prior knowledge into the choice of $f$ is indeed a meaningful extension, and we plan to explore this in future work.
>
> Due to the time constraints of the rebuttal phase, we include here a reference result using a linear combination of three uniform distributions for $f$ as a comparison:
>
> |      	  | $\mathcal{D}_E \to \mathcal{D}_D$ | $\mathcal{D}_G \to \mathcal{D}_D$ |
> |------------|------------|------|
> |GLSGE w/ Gau. | 6.21 | 7.30 |
> | GLSGE w/ Uni. | 6.02 | 6.58 |
> |DCUA  | 5.95 | 6.4 |
>
> These observations indicate that GLSGE has greater potential than what is shown in the current version.
>
> **_Q: Relationship between sections in Sec. 3._**
>
> We appreciate the reviewer’s comments regarding the organization of Sec. 3 and the suggestion to clarify the structure and flow. Here we provide a brief overview to clarify the writing logic of Sec. 3:
>
> -- Sec. 3.1: We reformulate the CDGE problem through a GLS perspective, identifying label shift and conditional shift as core challenges.
>
> -- Sec. 3.2: We introduce the GLS correction framework as a principled solution to CDGE. We then point out (in **lines 162–168**) that **existing DG and UDA methods can be viewed as partial realizations of this framework**—specifically, they focus on invariant representation learning while ignoring label shift correction. This perspective highlights why such methods may fail under significant label shift.
>
> -- Sec. 3.3-3.4: We propose two key components of our method tailored to the CDGE problem: Label Probability Distribution Estimation (Sec. 3.3), corresponding to Eq. (7), and Probability-Aware Conditional Alignment (Sec. 3.4), corresponding to Eq. (8).
>
> -- Sec. 3.5: We present a concrete implementation of the full GLS correction framework, combining the Eq. (20).
>
> We agree that the discussion of DG and UDA methods in Sec. 3.2 may initially appear to overlap with related work. However, we believe that **revisiting these methods after introducing our framework allows us to reinterpret existing approaches from a unified theoretical perspective, thereby revealing their inherent limitations and motivating our design choices more clearly.**
>
> We appreciate the suggestion, and in the final version we will improve the section transitions and **provide an overview at the beginning of Sec. 3** to better guide the reader through the structure.
>
> **_Q: How GLS correction methods have benefited other tasks and their value specifically in gaze estimation_**
>
> Thank you for raising this important point.
> The GLS framework was originally proposed in the study of domain shift to address scenarios where both label shift and conditional shift are present. We have provided a detailed discussion of related GLS theory and methods in **Sec. 2, lines 96–111**, where we note that most existing GLS approaches are designed for discrete classification tasks with a finite label space.
>
> In contrast, our work is among the first to explore the application of the GLS framework to CDGE problem with continuous labels. In Sec. 3.1, we analyze the domain gap in CDGE and argue that it naturally involves both label and conditional shifts. Therefore, leveraging the GLS perspective in this context is a natural and theoretically grounded choice. Moreover, this perspective allows us to reinterpret and unify prior CDGE methods and motivates a new class of solutions beyond existing paradigms.
>
> **_Q: An overview of the current state of gaze estimation research._**
>
> We appreciate the reviewer’s suggestion. An overview of the current state of gaze estimation research is provided in **Sec. 2, lines 82–95**. In that section, we summarize recent advances in CDGE, including both DG and UDA methods. Specifically, in **line 94-95**, we point out that they essentially focus on domain-invariant representation learning although differ in technology.
>
> In the final version, we will consider making this overview more explicit to better contextualize our contributions.

---

> > ### Comment · Reviewer_bWiS · 2025-08-08
> >
> > Thanks for the responses. After reviewing the rebuttal and considering the comments from the other reviewers, I am glad to maintain my original rating as borderline accept.

---

> > > ### Author Response · Authors · 2025-08-08
> > >
> > > Thank you for your positive feedback. Your comments are greatly appreciated and will be carefully incorporated in the final revision of our paper.

---

> ### Comment · Area_Chair_bybh · 2025-08-05
> **Please discuss rebuttal with authors**
>
> Please discuss rebuttal with authors

---

### Official Review · Reviewer_W2yU · 2025-06-30

**Clarity:** 1
**Significance:** 2
**Originality:** 3
**Rating:** 4
**Confidence:** 4

**Summary:**

This paper proposes to introduce GLS into the CDGE task, and designs a design that simultaneously utilizes GLS, UDA, and DG techniques to improve the performance of CDGE. The paper introduces a lot of theoretical analysis and derivations to introduce GLS into CDGE tasks.

**Questions:**

Alignment in Fig.1 and Probability-aware learning with ... in Fig.3 are in different colors, kind of confuse me, does this have a special meaning?

**Ethical Concerns:**

["NO or VERY MINOR ethics concerns only"]

**Final Justification:**

After carefully reviewing the authors' response and the comments from other reviewers, I find that some of my concerns have been addressed. Therefore, I am willing to slightly increase my score.

**Limitations:**

yes

**Quality:**

2

**Strengths And Weaknesses:**

**Strengths**

1.  The paper has a detailed ablation study, feature visualization and hyper-param visualization.
2.  The paper includes detailed propagation, which makes the paper more theoretically sound.

**Weaknesses**

1.  There are too many notations in the paper, which makes the paper not easy to follow.
2.  The results from other domains to EyeDiap are not good.
3.  Although the paper puts the enhancement of GLSGE by Res-18, Res50 and GazeTR, the enhancement is not easy to see compared with the enhancement of other methods on these backbones, if we can put the comparison of different methods on GazeTR+X (X is other methods), we can understand this paper better.
4.  Although the paper puts a lot of analysis of the introduced GLS theory, the results on the EyeDiap dataset are not so good, which makes me concern whether the theory is suitable enough for this estimation task.

**Minors**

Minors.

1.  L8: a → an
2.  Equations 2, 3, 8, 12, 19 lack the necessary punctuation

---

> ### Author Rebuttal · Authors · 2025-07-30
>
> Thank you for taking the time to review our paper and provide professional and constructive reviews. We are encouraged by the positive comments on the extensive experiments and the theoretical support. All suggestions will be carefully incorporated into final version.
>
> **_Q: There are too many notations in the paper._**
>
> We acknowledge the reviewer’s concern about the number of notations potentially impacting readability. While a relatively large set of symbols is necessary to ensure rigorous and precise method description, we took care to **define each notation clearly upon its first appearance.** We agree that streamlining the notation can further enhance accessibility. For the final version, we will make a concerted effort to simplify the notation without sacrificing rigor or clarity. For example, to **provide a comprehensive list of all notations in the appendix.**
>
> **_Q: Results on the EyeDiap dataset._**
>
> We acknowledge the reviewer's concern about GLS performance on EyeDiap. While the absolute performance is not SOTA, we emphasize two key points:
>
> --**Validity.** GLS demonstrates significant improvement over the backbone model (e.g., error reduction of 31.2% in $\mathcal{D}_E \to \mathcal{D}_D$ with backbone Res-18), indicating its effectiveness for the EyeDiap dataset.
>
> --**Implementation Flexibility.** The current implementation of GLSGE is only **an instantiation of the general framework** (e.g., truncated 2D Gaussian for label distribution). As our core goal is to present the GLS perspective for CDGE, we intentionally **avoided too much dataset-specific tuning in evaluation.** As shown below,
> simply tuning the truncated 2D Gaussian to linear combination of three uniform distributions can improve the performance in EyeDiap with Res-18, making it comparable to SOTA methods.
>
> |      	  | $\mathcal{D}_E \to \mathcal{D}_D$ | $\mathcal{D}_G \to \mathcal{D}_D$ |
> |------------|------------|------|
> |GLSGE w/ Gau. | 6.21 | 7.30 |
> | GLSGE w/ Uni. | 6.02 | 6.58 |
> |DCUA  | 5.95 | 6.4 |
>
> These observations indicate that **GLSGE has greater potential than what is shown in the current version.** We will emphasize this point more clearly in the final version of the paper to better convey the strength and flexibility of the framework.
>
> **_Q: Comparison based on GazeTR._**
>
> Thank you for your valuable suggestion. In **Tab. 2 & 3**, we compare GLSGE with other methods using ResNet-18 and ResNet-50 as backbones. The results show that GLSGE brings more significant improvements to these backbones compared to other methods, which demonstrates the effectiveness and general applicability of our approach.
>
> As for GazeTR, due to the lack of publicly available GazeTR-based CDGE methods, we present the results in Table 3 in the form of plug-play studies. The results still indicate clear performance gains, suggesting that GLSGE is compatible with transformer-based architectures as well.
>
> In the final version, we plan to reproduce and incorporate some representative plug-and-play methods with the GazeTR backbone (e.g., GazeTR+PnP-GA), to provide a more comprehensive comparison. We believe this will further enhance the clarity and completeness of our experimental analysis.
>
> **_Q: Colors of "Alignment" in Fig. 1 and "Probability-aware learning with \omega" in Fig. 3._**
>
> Thank you for pointing this out. The color choices in Fig. 1 and 3 were intentionally designed to provide more intuitive visual cues, though we understand they may have caused confusion.
>
> In Fig. 1, the orange solid line represents the GLS method, and the black solid line represents the UDA method. **The color of the "Alignment" component is split into orange and black to indicate that both GLS and UDA perform distribution alignment.**
>
> In Fig. 3, the text color reflects the component each learning objective is associated with: red for importance reweighting, blue for invariant representation learning, and purple for predictor learning. Since the weight function ω(y) is learned from the importance reweighting module, we use red for the term ω(y). At the same time, ω(y) also plays a role in both invariant representation learning and predictor learning. Thus, **the term probability-aware learning with ω is colored in a combination of blue and purple to reflect its joint influence.**
>
> We will clarify this design in the caption of the figures in the final version to avoid misunderstanding.
>
> **_Q: Typo._**
>
> We sincerely appreciate the reviewer's careful reading. All reported typos will be corrected in the final manuscript, and we will conduct a full proofread to ensure textual accuracy throughout the paper.

---

> > ### Comment · Reviewer_W2yU · 2025-08-07
> >
> > Thank you for the authors' response. My previous concerns regarding the experimental performance, the lack of certain comparisons, and some missing details have been partially addressed. I strongly encourage the authors to incorporate these improvements into the final version, as they are crucial for the completeness and reproducibility of the work. Therefore, I am willing to slightly raise my final score.

---

> > > ### Author Response · Authors · 2025-08-08
> > >
> > > Thank you very much for your positive feedback and for raising the score. Your comments are greatly appreciated and will be carefully incorporated in the final revision of our paper to improve the completeness and reproducibility.

---

> ### Comment · Area_Chair_bybh · 2025-08-05
> **Please discuss rebuttal with authors**
>
> Please discuss rebuttal with authors

---

> ### Author Response · Authors · 2025-08-07
>
> Thank you again for your time and thoughtful feedback. We have provided detailed responses and additional experiments based on your comments. If possible, we would be very grateful to know whether you have any further questions or feedback, as your response is very important for us at this stage.

---

### Note · Authors · 2025-08-12

We sincerely thank all reviewers and the AC for your effort and constructive feedback, which have been invaluable for improving this work. Here we briefly summarize the rebuttal:

**Strengths noted by reviewers:**

(1) Novel and theoretically sound perspective for domain adaptation in gaze estimation.

(2) Comprehensive experiments.

(3) Clear explanation and readability.

**Main concerns raised**:

(1) Methodology rationale (why GLS is crucial for CDGE, label distribution modeling choice, and conditional distribution decomposition).

(2) Experimental settings (performance on EyeDiap, number of target-domain samples).

(3) Presentation details (table formatting, figure clarity, and broader applicability).

**Rebuttal highlights:**

We clarified that

-- CDGE's domain gap stems from both label and conditional shift, often overlooked by prior methods.

-- our GLS correction framework is general for continuous-label cross-domain problems, but our contribution goes beyond proposing the framework: we designed a task-specific implementation that makes GLS effective for CDGE, a step essential for adapting GLS to other tasks.

-- our conditional distribution decomposition offers a more interpretable formulation for CDGE.

We justified the bivariate truncated Gaussian choice by outlining essential modeling properties and noting alternative options for different applications.

Additional experiments showed strong performance on EyeDiap and confirmed effectiveness even with fewer target samples, while explaining the practicality of using slightly more unlabeled samples.

Presentation and clarity issues were addressed, and we discussed connections to other tasks, expanding the potential impact of this work.

**We are pleased that all reviewers indicated their main concerns were resolved and reached a consensus on a positive and supportive attitude toward the paper. We appreciate the reviewers’ recognition of the paper’s novelty, methodological soundness, and practical relevance, and look forward to the opportunity for this work to contribute to the community.**

---

### Decision · Program_Chairs · 2025-09-17

**Decision:**

Accept (poster)

**Comment:**

The paper proposes a novel Generalized Label Shift framework for cross-domain gaze estimation. The authors effectively addressed reviewer concerns regarding method rationale, experimental design (including performance on EyeDiap and the use of target data), and presentation clarity, through detailed explanations and new experiments. All reviewers acknowledged that their concerns were adequately resolved and maintained a positive, borderline accept score. Therefore, I recommend acceptance.